# A US case-control study to estimate infant group B streptococcal disease serological thresholds of risk-reduction

Julia C. Rhodes [1,19] ✉, Rebecca Kahn[1,2,19], Shanna Bolcen [1], Nong Shang[1], Yunmi Chung [1], Monica M. Farley [3], Amber N. Britton[3], Ashley E. Moore [4], Stepy Thomas[3], Joelle Nadle[5], Laura B. Amsden[6], Jacek Skarbinski[6], Kristi R. VanWinden [7], Meghan Barnes[8], Kerianne Engesser [9], Patricia Ferrieri[10], AmberJean P. Hansen[11], Lee H. Harrison [12], Laura Jeffrey[12], Jessica L. Nyholm[10], Sean T. O'Leary[13], Courtney Olson-Chen[14], Jemma V. Rowlands [9], Shannon A. Seopaul [12], Ann R. Thomas[15], Htet Htet N. Wrigley [16], Lesley McGee [1], Sundaram Ajay Vishwanathan [1], Fahmina Akhter[1], Bailey Alston [17], Lily Tao Jia[1], Yikun Li [18], Palak Y. Patel [1], Joy Rivers[1], Jessica E. Southwell [18], Theresa Tran[1], Panagiotis Maniatis [1] & Stephanie J. Schrag [1] ✉

Maternal vaccines to prevent infant Group B *Streptococcus* (GBS) disease have progressed through phase II development and may be licensed based on immunologic endpoints, which have yet to be approved by regulatory authorities. Here we present a multistate case control study to characterize the relationship between serotype-specific anti-capsular polysaccharide (CPS) immunoglobulin G concentrations near birth and infant GBS disease risk reduction. Antibody concentration distributions are significantly lower for cases (n = 643) than controls (n = 2801) and serologic thresholds varied by serotype and age at onset, with 80% serotype-specific protective thresholds ranging from 0.52 to 2.49 mcg/mL for early-onset disease (EOD; <7 days old) and 0.02 to 0.14 mcg/mL for late-onset disease (LOD; 7-89 days old). Our study provides the most robust data to date that protection thresholds vary by serotype and are notably lower for LOD than EOD, thereby informing potential serological endpoints for phase III trials evaluating CPS-based maternal GBS vaccine candidates.

Maternal vaccination is a proven strategy to protect infants from infections during the neonatal period when the immune system is not fully developed and disease risk is at a lifetime high[1,2]. Expansion of maternal vaccines to target bacterial causes of neonatal sepsis could help prevent the estimated annual 3 million neonatal sepsis cases and 350,000 deaths[3]. Group B *Streptococcus* (GBS) is a leading cause of neonatal sepsis, resulting in ~400,000 invasive disease cases, 37,000 infants with long-term neurodevelopmental injuries, and over 90,000

infant deaths annually worldwide; case fatality rates among infants with onset in the first week of life (early-onset disease or EOD, the majority of which occurs on the day of birth) range from 0.06 to 0.23 by global region, whereas case fatality rates among infants with onset on days 7–89 of life (late-onset disease or LOD) are lower and less variable (0.06–0.10)[4]. Intrapartum antibiotic prophylaxis (IAP) based on antenatal screening for GBS colonization is currently the only available prevention strategy. While highly effective against EOD[5,6], IAP

is often neither available nor practical in resource-limited settings. IAP also does not prevent LOD, which accounts for an increasing proportion of infant GBS disease[7–10].

Maternal GBS vaccine candidates have progressed through phase-II trials, demonstrating immunogenicity and safety in pregnant women[10–15]. At least two products, GBS6, a conjugate product based on the capsular polysaccharides (CPS) of the six most common infant GBS serotypes, and GBS-NN/NN2, a vaccine based on GBS surface proteins, have received fast-track designations[16,17]. Given the sample size requirements and other challenges with traditional disease endpoint phase-III trials for GBS vaccine candidates[18], regulators have expressed openness to vaccine licensure based on immunologic endpoints[16,19], which have yet to be approved by regulatory authorities. Because maternal antibody transfer ratios to newborns can be affected by factors such as gestational age at delivery, immunological endpoints based on cord blood or newborn blood at birth have been the area of focus[20]. Previously conducted sero-epidemiological studies have documented lower anti-GBS CPS immunoglobulin G antibody concentrations (anti-CPS IgG) in cord blood from infant GBS cases compared to controls and proposed potential protective thresholds[21,22]. Recently published studies conducted in South Africa and Finland used updated, standardized assays and estimated protective thresholds based on infant anti-CPS IgG, but small sample sizes limited stratification by serotype or by EOD vs LOD[8,10].

We conducted the largest sero-epidemiologic study to date, leveraging countrywide, state-administered newborn screening programs and the Centers for Disease Control and Prevention's (CDC) Active Bacterial Core surveillance (ABCs) for invasive GBS disease[23]. Here we used a case-control analysis to characterize the relationship between newborn serotype-specific anti-CPS IgG antibody concentrations and disease risk reduction to estimate protective thresholds for EOD and LOD to inform immunologic endpoints for maternal GBS vaccine trials.

## Results

We enrolled 643 cases (268 EOD; 375 LOD) and 2801controls (Fig. 1). Cases were defined as infants <90 days old with isolation of GBS from a normally sterile site. Controls were defined as infants born to GBS-colonized women based on routine antenatal or intrapartum screening who did not develop invasive GBS disease in the first 90 days of life. Cases and controls differed on a number of maternal, intrapartum, and newborn characteristics in line with the epidemiology of infant GBS disease (Tables 1–2). Over a third of cases (39%; 251/643) were born preterm, including 28% (181/643) before 34 weeks of gestation, in contrast to 12% (344/2801) and 4% (100/2801) of controls, respectively. Intrapartum fever and diagnosis of intraamniotic infection were more common among EOD cases than among LOD cases or controls. A large

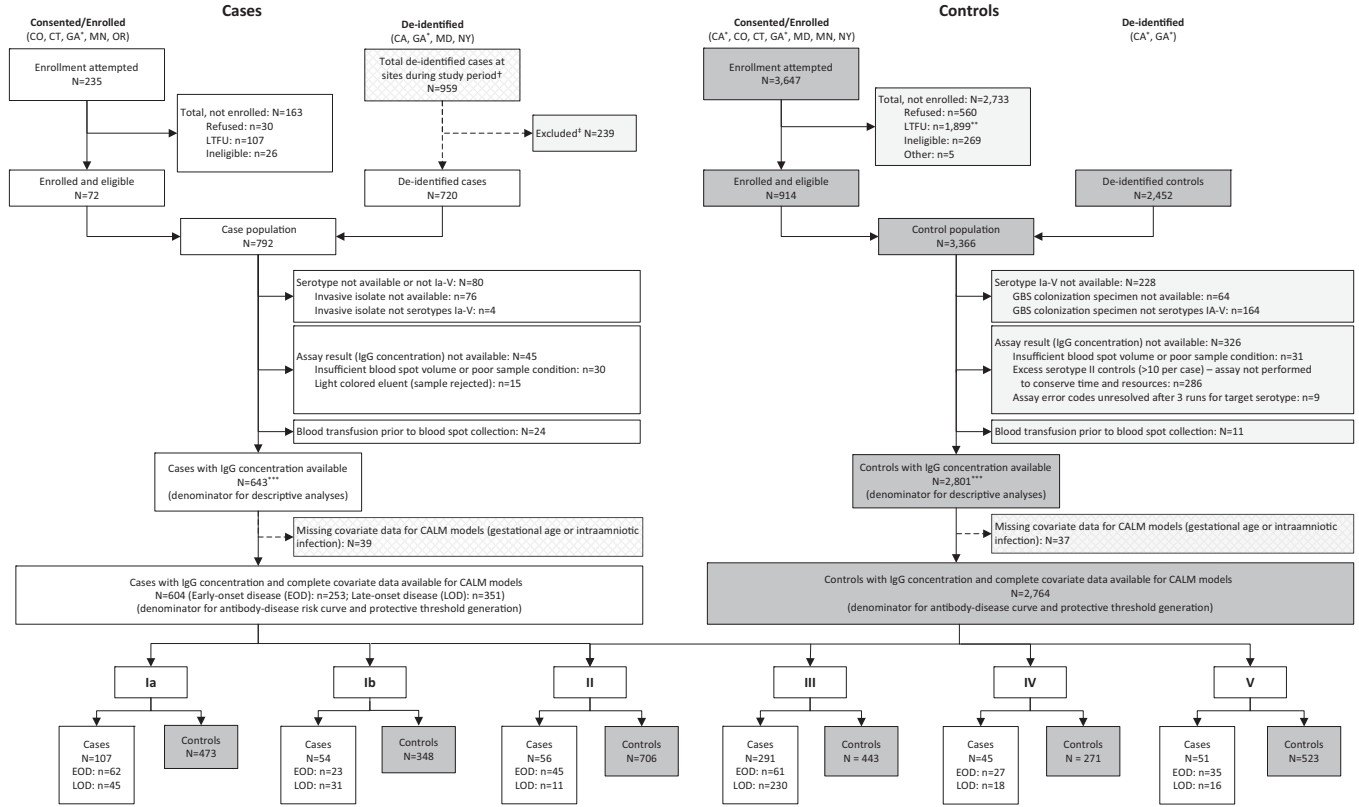

**Fig. 1 | Study population enrollment flow diagram.** Flow chart showing enrollment process for cases and controls. Final sample sizes for curve and protective threshold generation, disaggregated by early-onset disease and late-onset disease for each serotype, are shown at the bottom. *Cases and controls were initially enrolled only after individual-level consent. Subsequent to approval for a waiver of individual-level consent, cases and controls were enrolled as de-identified samples. †Time periods for the inclusion of de-identified samples differed by site, depending on site-specific blood spot retention policies - CA 2015–2022; GA 2018–2022; MD 2013–2022; NY 2010–2022. ‡Sites reported that cases were not included because of blood spot unavailability; however, sites were not permitted to disclose individual-level reasons for excluding ABCs cases in accordance with the study de-identification strategy. Most of the excluded cases (178/239) came from one site

with a short blood spot retention period, and spots were sometimes discarded prior to collection for study purposes. **Includes n = 588 GBS colonized women who did not respond to a single mailer regarding potential study participation per site ethics board restrictions on enrollment attempts. ***Consented enrollment was used for 61/643 (9.4%) of cases and 727/2801 (26%) of controls. Control serotype could not be determined until after enrollment and processing of the maternal GBS screening specimens at CDC. As a result, the number of controls per case ranges from 1.5 for serotype III to 13.1 for serotype II due to differences in the serotype distribution of GBS isolates from infant cases vs the serotype distribution of specimens from GBS colonized pregnant women. LTFU lost to follow-up, CALM covariate-adjusted logit model.

## Table 1 | Maternal and intrapartum characteristics by case and control status

| | Early-onset cases: (0–6 days) n (col %) | Late-onset cases: (7–89 days) n (col %) | All Cases n (col %) | Controls n (col %) |
|---|---|---|---|---|
| Total | 268 (100) | 375 (100) | 643 (100) | 2801 (100) |
| **Maternal characteristics** | | | | |
| Age at delivery (years) median (min, max) | 28 (15–58) | 29 (14–44) | 29 (14–58) | 31 (14–52) |
| <18 | 6 (2) | 12 (3) | 18 (3) | 40 (1) |
| 18–24 | 73 (27) | 113 (30) | 186 (29) | 461 (16) |
| 25–34 | 141 (53) | 180 (48) | 321 (50) | 1525 (54) |
| ≥35 | 46 (17) | 61 (16) | 107 (17) | 762 (27) |
| Missing | 2 (<1) | 9 (2) | 11 (2) | 13 (<1) |
| Study site | | | | |
| CA | 29 (11) | 53 (14) | 82 (13) | 1080 (39) |
| CO[a] | NS | NS | NS | 26 (<1) |
| CT | NS | NS | NS | 78 (3) |
| GA | 85 (32) | 87 (23) | 172 (27) | 1373 (49) |
| MD | 95 (35) | 116 (31) | 211 (33) | 105 (4) |
| MN | 12 (4) | 32 (9) | 44 (7) | 19 (<1) |
| NY | 41 (15) | 72 (19) | 113 (18) | 120 (4) |
| OR | NS | NS | NS | 0 |
| Maternal medical conditions[b,c] | N = 117 | N = 125 | N = 242 | N = 2801 |
| Obesity | 17 (15) | 12 (10) | 29 (12) | 438 (16) |
| Asthma | 13 (11) | 15 (12) | 28 (12) | 374 (13) |
| Diabetes mellitus | 4 (3) | 3 (2) | 7 (3) | 122 (4) |
| Chronic skin breakdown | 0 | 0 | 0 | 53 (2) |
| Immunosuppressive therapy (steroids, etc.) | 0 | 2 (2) | 2 (<1) | 19 (1) |
| HIV infection[d] | 0 | 0 | 0 | 20 (<1) |
| Complement deficiency | 0 | 0 | 0 | 0 |
| Insurance type | | | | |
| Private | 172 (64) | 235 (63) | 407 (63) | 1685 (60) |
| Medicaid or other gov't/state assistance | 76 (28) | 118 (31) | 194 (30) | 1022 (36) |
| Uninsured | 12 (4) | 8 (2) | 20 (3) | 26 (<1) |
| Other | 2 (<1) | 5 (1) | 7 (1) | 37 (1) |
| Missing | 6 (2) | 9 (2) | 15 (2) | 31 (1) |
| Number of prior pregnancies[e,f] | N = 189 | N = 261 | N = 450 | N = 2801 |
| 0 | 54 (29) | 79 (30) | 133 (30) | 1216 (43) |
| 1 | 62 (33) | 61 (23) | 123 (27) | 837 (30) |
| 2 or more | 55 (29) | 90 (34) | 145 (32) | 711 (25) |
| Missing | 18 (10) | 31 (12) | 49 (11) | 37 (1) |
| Number of antenatal visits during this pregnancy | | | | |
| 1–4[g] | 20 (7) | 33 (9) | 53 (8) | 153 (5) |
| 5–7 | 44 (16) | 54 (14) | 98 (15) | 309 (11) |
| 8 or more | 121 (45) | 176 (47) | 297 (46) | 2159 (77) |
| Missing | 83 (31) | 112 (30) | 195 (30) | 180 (6) |
| GBS bacteriuria during this pregnancy | | | | |
| Yes | 32 (12) | 36 (10) | 68 (11) | 135 (5) |
| No | 170 (63) | 241 (64) | 411 (64) | 2618 (93) |
| Missing | 66 (25) | 98 (26) | 164 (26) | 48 (2) |
| Antenatal GBS screening during this pregnancy | | | | |
| Yes | 180 (67) | 250 (67) | 430 (67) | 2801 (100) |
| No | 57 (21) | 69 (18) | 126 (20) | |
| Missing | 31 (12) | 56 (15) | 87 (14) | |
| If yes, was screening performed prior to hospital admission for delivery? | 155/180 (86) | 189/250 (76) | 344/430 (80) | 2630/2801 (94) |
| If yes, was screening performed during hospital admission for delivery? | 30/180 (17) | 67/250 (27) | 97/430 (23) | 231/2801 (8) |
| If yes, screening test result was positive | 56/180 (31) | 138/250 (55) | 194/430 (45) | 2801 (100) |
| If yes, days between positive GBS screening and birth, Median (min–max)[h] | N = 50 20 (<1–151) | N = 122 16 (<1–64) | N = 172 17 (<1–151) | N = 2785 20 (<1–143) |

## Table 1 (continued) | Maternal and intrapartum characteristics by case and control status

| | Early-onset cases: (0–6 days) n (col %) | Late-onset cases: (7–89 days) n (col %) | All Cases n (col %) | Controls n (col %) |
|---|---|---|---|---|
| **Intrapartum characteristics** | | | | |
| Duration of membrane rupture (hours) | | | | |
| Median (min–max)[h] | 10 (<1–709) | 3 (<1–376) | 6 (<1–709) | 4 (<1–1251) |
| Missing n (%) | 47 (18) | 85 (23) | 132 (21) | 189 (7) |
| Duration of membrane rupture ≥18 hours | | | | |
| Yes | 61 (23) | 53 (14) | 114 (18) | 330 (12) |
| No | 190 (71) | 283 (75) | 473 (74) | 2351 (84) |
| Missing | 17 (6) | 39 (10) | 56 (9) | 120 (4) |
| Preterm premature rupture of membranes–PPROM[g] (among those with gestational age <37 weeks) | 46/74 (62) | 57/177 (32) | 103/251 (41) | 166/344 (48) |
| Intrapartum fever: T > 100.4 F or 38.0 C | | | | |
| Yes | 54 (20) | 12 (3) | 66 (10) | 162 (6) |
| No | 196 (73) | 339 (90) | 535 (83) | 2602 (93) |
| Missing | 18 (7) | 24 (6) | 42 (7) | 37 (1) |
| Intraamniotic infection–noted or suspected[c] | | | | |
| Yes | 88 (33) | 26 (7) | 114 (18) | 311 (11) |
| No | 164 (61) | 325 (87) | 489 (76) | 2453 (88) |
| Missing | 16 (6) | 24 (6) | 40 (6) | 37 (1) |
| Evidence of intraamniotic infection: intrapartum fever OR intraamniotic infection | | | | |
| Yes | 97 (36) | 29 (8) | 126 (20) | 355 (13) |
| No | 156 (58) | 322 (86) | 478 (74) | 2409 (86) |
| Missing | 15 (6) | 24 (6) | 39 (6) | 37 (1) |
| Intrapartum antibiotic prophylaxis | | | | |
| None | 148 (55) | 131 (35) | 279 (43) | 302 (11) |
| Some, but inadequate (< 4 hrs or not a beta-lactam antibiotic) | 71 (26) | 109 (29) | 180 (28) | 694 (25) |
| Adequate (≥4 hours and beta-lactam antibiotic) | 33 (12) | 112 (30) | 145 (23) | 1772 (63) |
| Missing | 16 (6) | 23 (6) | 39 (6) | 33 (1) |
| Delivery type by Cesarean section | | | | |
| Vaginal delivery | 179 (67) | 226 (60) | 405 (63) | 1898 (68) |
| C–section | 88 (33) | 138 (37) | 226 (35) | 879 (31) |
| Missing | 1 (<1) | 11 (3) | 12 (2) | 24 (<1) |
| If C-section, did membranes rupture prior to C-section? | N = 88 | N = 138 | N = 226 | N = 879 |
| Yes | 74 (84) | 52 (38) | 126 (56) | 372 (42) |
| No | 13 (15) | 73 (53) | 86 (38) | 477 (54) |
| Missing | 1 (1) | 13 (9) | 14 (6) | 30 (3) |

[a]Data not shown (NS) reflect small sample sizes (< 10) with exact numbers suppressed to protect participant confidentiality.
[b]Select conditions are presented because they were common among our study population (e.g., obesity) or related to immune function (e.g., HIV infection). Case data limited to 2020–2022; not routinely collected pre-2020.
[c]Marked as 'yes' if noted in the hospital chart.
[d]HIV status was not reported by one site.
[e]Number of prior pregnancies was added to the ABCs case report form in 2018. Sites re-abstracted older records, with the exceptions of MD and NY.
[f]A previous infant with GBS disease was noted in the hospital chart for 7/498 cases and 3/2566 controls with data available.
[g]Includes 1 control with no antenatal visits.
[h]Calculated among those with data available.

majority of controls received IAP (88%; 2466/2801), though fewer (63%; 1772/2801) received adequate IAP, defined as 4 or more hours of a beta-lactam intravenous antibiotic before delivery. Just over half of the cases received IAP (51%; 325/643), and fewer received adequate IAP: 12% of EOD cases and 30% of LOD cases.

Remnant routine newborn screening dried blood spot (DBS) samples allowed for antibody concentration measurement using a 6-plex (Ia, Ib, II, III, IV, and V) anti-CPS IgG direct Luminex immunoassay (dLIA). Overall, 88% (3016/3444) of DBS were collected within 48 hours of birth (498/643 or 77% of case DBS; 2518/2801 or 90% of control

**Table 2 | Newborn and blood spot sample characteristics by case and control status**

| | Early-onset (0–6 days) cases: n (col %) | Late-onset (7–89 days) cases: n (col %) | All cases n (col %) | Controls n (col %) |
|---|---|---|---|---|
| Total | 268 (100) | 375 (100) | 643 (100) | 2801 (100) |
| **Newborn characteristics** | | | | |
| Gestational age (weeks) | | | | |
| <34 | 54 (20) | 127 (34) | 181 (28) | 100 (4) |
| 34–36 | 20 (7) | 50 (13) | 70 (11) | 244 (9) |
| ≥37 | 193 (72) | 197 (53) | 390 (61) | 2449 (87) |
| Missing | 1 (<1) | 1 (<1) | 2 (<1) | 8 (<1) |
| Birth weight in grams, | | | | |
| Median (min–max) | 3150 (455–4740) | 2610 (360–4770) | 2853 (360–4770) | 3220 (445–6650) |
| Missing n (%) | 0 (0) | 14 (4) | 14 (2) | 30 (1) |
| Duration of birth hospitalization (days) | | | | |
| Median (min–max) | 11 (<1–141) | 4 (<1–214) | 10 (0–214) | 2 (0–348) |
| Missing n (%) | 11 (4) | 26 (7) | 37 (6) | 41 (1) |
| Receipt of antibiotics during birth hospitalization[a] | N = 117 | N = 125 | N = 242 | N = 2799 |
| Yes | 99 (85) | 57 (46) | 156 (64) | 383 (14) |
| No | 9 (8) | 54 (43) | 63 (26) | 2399 (86) |
| Missing | 9 (8) | 14 (11) | 23 (10) | 17 (<1) |
| Race and ethnicity | | | | |
| Hispanic or Latino | 32 (12) | 53 (14) | 85 (13) | 497 (18) |
| Non-Hispanic (NH) Black | 102 (38) | 138 (37) | 240 (37) | 1136 (41) |
| NH Asian or Pacific Islander | 15 (6) | 28 (7) | 43 (7) | 418 (15) |
| NH American Indian or Alaska Native | 0 | 0 | 0 | 2 (<1) |
| NH White | 102 (38) | 146 (39) | 248 (39) | 649 (23) |
| Missing | 17 (6) | 10 (3) | 27 (4) | 99 (4) |
| Sex | | | | |
| Male | 139 (52) | 189 (50) | 328 (51) | 1427 (51) |
| Female | 127 (47) | 182 (49) | 309 (48) | 1373 (49) |
| Missing | 2 (<1) | 4 (1) | 6 (<1) | 1 (<1) |
| Year of birth | | | | |
| 2010–2012 | 13 (5) | 18 (5) | 31 (5) | 0 (0) |
| 2013–2017 | 84 (31) | 139 (37) | 223 (35) | 0 (0) |
| 2018–2022[b] | 171 (64) | 218 (58) | 389 (60) | 2801 (100) |
| Age at GBS illness onset (cases only) | | | | – |
| Median (days) (IQR) | 0 (0–0) | 31 (18–49) | 13 (0–36) | – |
| Syndrome (cases only) | | | | – |
| Bacteremia without focus | 212 (79) | 199 (53) | 411 (64) | – |
| Meningitis | 34 (13) | 140 (37) | 174 (27) | – |
| Other[c] | 22 (8) | 36 (10) | 58 (9) | – |
| Outcome of GBS illness (cases only) | | | | – |

**Table 2 (continued) | Newborn and blood spot sample characteristics by case and control status**

| | Early-onset (0–6 days) cases: n (col %) | Late-onset (7–89 days) cases: n (col %) | All cases n (col %) | Controls n (col %) |
|---|---|---|---|---|
| Survived | 251 (94) | 347 (93) | 598 (93) | – |
| Died | 16 (6) | 27 (7) | 43 (7) | – |
| Missing | 1 (<1) | 1 (<1) | 2 (<1) | – |
| **Blood spot characteristics** | | | | |
| Age of the baby when DBS was collected | | | | |
| Median (days) (IQR) | 1 (0–2) | 1 (1–2) | 1 (1–2) | 1 (1–1) |
| Missing n (%) | 0 (0) | 1 (<1) | 1 (<1) | 9 (<1) |
| Age of baby when DBS was collected | | | | |
| <24 hours (0 day)[d] | 125 (47) | 102 (27) | 227 (35) | 633 (23) |
| 24–<48 hours (1 day) | 73 (27) | 198 (53) | 271 (42) | 1,885 (67) |
| 48–<72 hours (2 days) | 36 (13) | 44 (12) | 80 (12) | 198 (7) |
| 72–<96 hours (3 days) | 12 (4) | 5 (1) | 17 (3) | 13 (<1) |
| 96 hours or more (4+ days) | 22 (8) | 25 (7) | 47 (7) | 63 (2) |
| Missing | 0 | 1 (<1) | 1 (<1) | 9 (<1) |
| Timing of DBS collection relative to onset of GBS illness (cases only) | | | | |
| Before or on the same day as the case–defining culture | 123 (46) | 370 (99) | 493 (77) | – |
| 1–3 days after | 120 (45) | 1 (<1) | 121 (19) | – |
| >3 days after | 25 (9) | 3 (<1) | 28 (4) | – |
| Missing | 0 | 1 (<1) | 1 (<1) | – |

[a]Case data limited to 2020–2022; not routinely collected pre-2020.
[b]Enrollment ended 31 Dec 2022, but six controls were born in early 2023 after their mothers were identified as GBS colonized and enrolled near the end of 2022.
[c]Includes other invasive syndromes, the most common of which were cellulitis, septic shock, and bacteremic pneumonia.
[d]Number of days used if hours were not available due to missing time of collection or time of birth.

DBS). Among cases, 4% (28/643) of DBS were collected more than 3 days (72 hours) after onset of disease (Table 2).

For some serotypes, the proportion of specimens below the LLOQ (lower limit of quantification) was higher among specimens collected before 2013 (>~4000 days before antibody testing) compared to more recent specimens, potentially consistent with antibody degradation after long storage times (Supplementary Figs. 1–3). However, this finding was not consistent across serotypes (notable for Ia, but minimal for III, IV, and V) and declines in antibody concentrations were not observed when specimens were restricted to those with measurable concentrations (Supplementary Figs. 1–3). We therefore included all specimens in the primary analysis and assessed the effects of long storage times in a sensitivity analysis.

We also did not observe any clear trends in the relationship between antibody concentrations and timing of DBS collection relative to disease onset (Supplementary figs. 4–5) and so did not restrict the primary analytic sample size based on timing of DBS collection.

Antibody concentration distributions differed significantly between cases and controls for the majority of target serotypes and disease onset strata, as evidenced by bee swarm and box plots (Fig. 2), receiver operating characteristic (ROC) curves (Supplementary Fig. 6), area under the curve (AUCs), and Wilcoxon rank sum tests (Supplementary Table 1). Differences were approaching

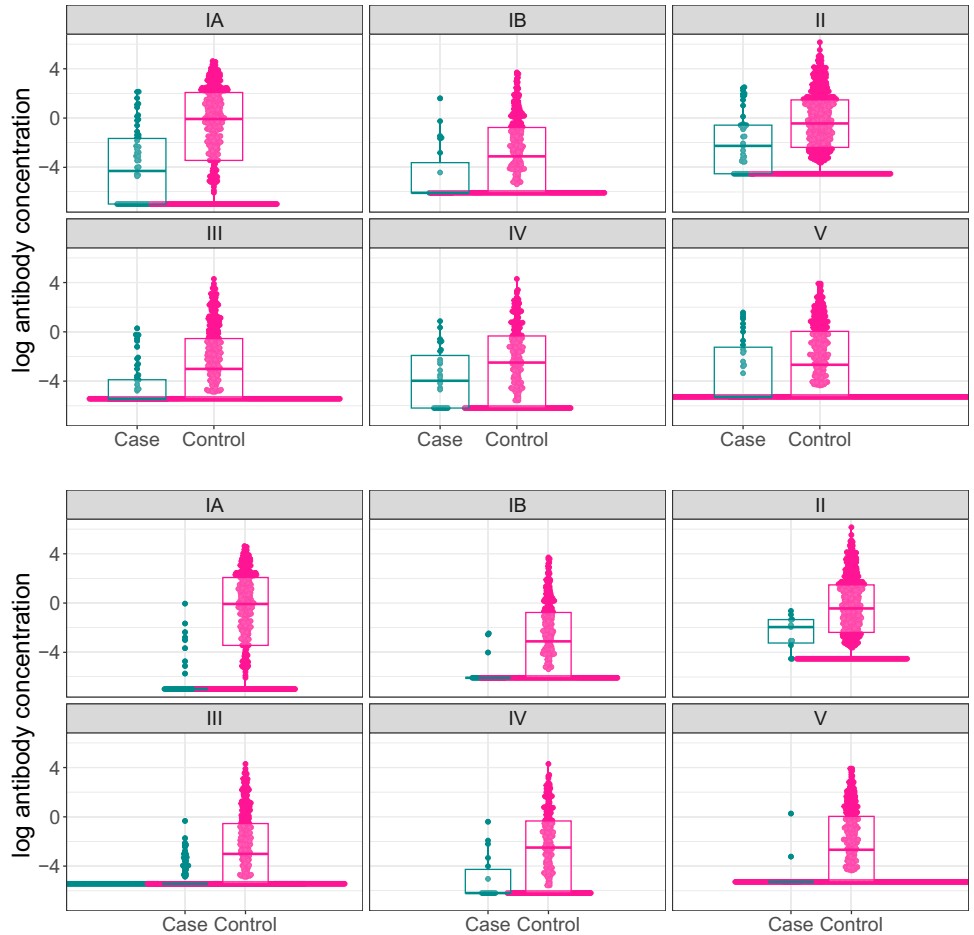

**Fig. 2 | Log anti-capsular polysaccharide IgG antibody concentrations by case and control status.** Individual log anti-capsular polysaccharide IgG antibody concentrations (mcg/mL) (points) and distributions (box plots) for cases and controls are shown for each serotype (panels) for early-onset disease (EOD) (top panel) and late-onset disease (LOD) (bottom panel). The points indicate individual samples; the bounds of the box from the boxplots indicate the 25th and 75th percentiles, and the line shows the median (50th percentile) value. EOD: % above the lower limit of quantification for cases: IA−55% (34/62) for cases and 82% (390/473) for controls, IB−30% (7/23) for cases and 67% (234/348) for controls, II−69% (31/45) for cases and 87% (614/706) for controls, III−34% (21/61) for cases and 63% (279/443 for controls, IV−63% (17/27) for cases and 74% (201/271) for controls, V−46% (16/35) for cases and 63% (329/523) for controls. LOD: % above the lower limit of quantification for cases: IA−20% (9/45) for cases and 82% (390/473) for controls, IB−10% (3/31) for cases and 67% (234/348) for controls, II−82% (9/11) for cases and 87% (614/706) for controls, III−14% (32/230) for cases and 63% (278/443 for controls, IV−33% (6/18) for cases and 74% (201/271) for controls, V−13% (2/16) for cases and 63% (329/523) for controls) (See Supplementary table 2 for more details).

statistical significance for serotype IV EOD ($p = 0.05$) and serotype V EOD ($p = 0.08$).

When records with missing covariate values were dropped, 604 cases (253 EOD and 351 LOD) and 2764 controls were available for our primary analyses. While a substantial proportion of both cases and controls had values below the LLOQ (Fig. 2, Supplementary Table 2), the proportion of cases with antibody concentrations above the LLOQ was higher among controls and generally higher among EOD cases than LOD cases (Fig. 2, Supplementary Table 2). All serotype and age-at-onset combinations met sample size requirements for curve generation except serotypes Ib LOD and V LOD (Supplementary Table 2).

We found no evidence of covariate interactions in models that converged when interaction terms were included. We generated curves to characterize the relationship between newborn serotype-specific anti-CPS IgG antibody concentrations and disease risk reduction. Curves and protective thresholds generated with the Covariate Adjusted Logit Model (CALM) method, our primary method, varied by serotype and age at onset (Figs. 3, 4, Table 3), with 80% protective thresholds ranging from 0.52 to 2.49 mcg/mL for EOD and 0.02 to 0.14 mcg/mL for LOD. Zero risk

thresholds ranged from 1.53 to 12.91 mcg/mL for EOD and from 0.10 to 1.76 mcg/mL for LOD. There was substantial heterogeneity by serotype, with serotypes II, IV, and V having higher protective thresholds. For serotypes where both EOD and LOD curves were generated, EOD protective thresholds were consistently higher than LOD protective thresholds. Confidence bounds were tighter around LOD than EOD protective thresholds, with 90% risk reduction protective thresholds generally having wider bounds than 80% and 75%. For all serotype and age at onset strata except serotype II EOD, models converged to a single curve regardless of starting parameter values. For serotype II EOD, the model converged to two different curve shapes depending on starting parameter values; for one curve relative disease risk decreased moderately as antibody concentrations increased; a less common result (out of 100 runs of randomly selected starting parameter values, 7% yielded this result) was a curve with relative disease risk at 100% below a threshold antibody concentration and at 0% above that threshold value (i.e. a step function). This rarer curve had a marginally higher maximum likelihood value (i.e., statistically it fit the data better) and was therefore selected as the final curve for this stratum.

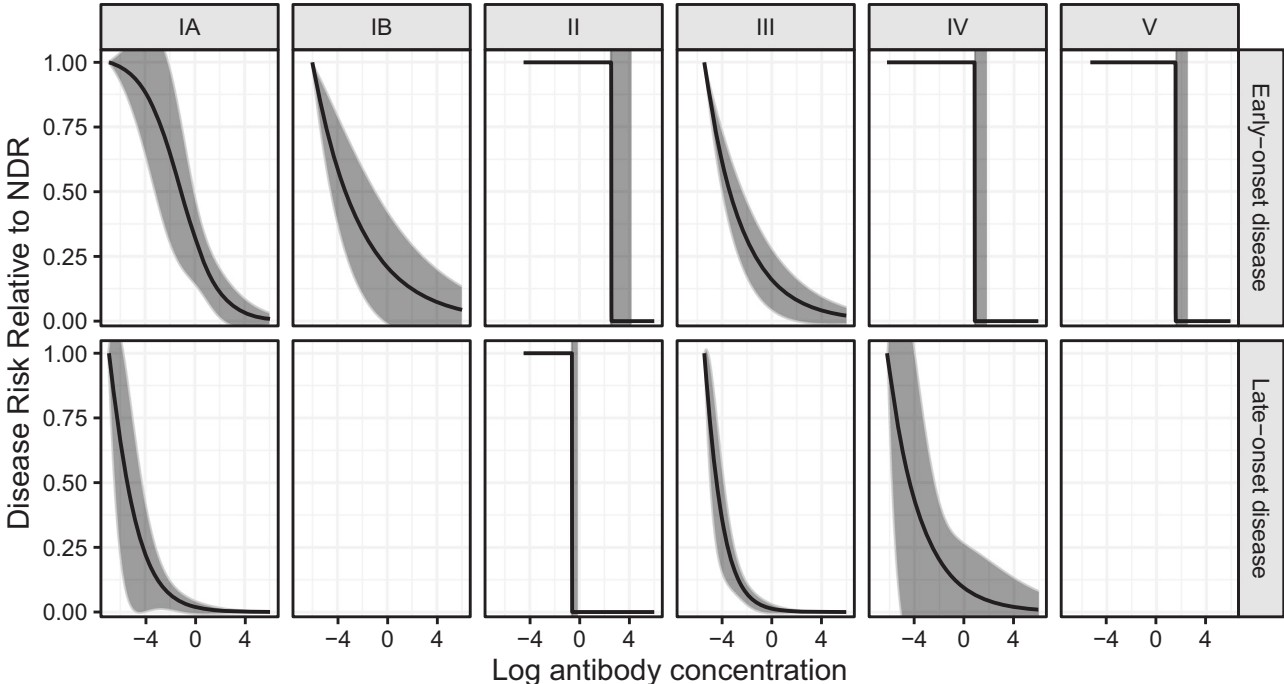

**Fig. 3 | Risk curves for anti-capsular polysaccharide IgG antibody concentrations and 95% confidence intervals.** Risk curves were generated using the covariate-adjusted logit model (CALM), adjusted for gestational age, intraamniotic infection, and study site, and are shown for early-onset disease (EOD) (top row) and late-onset disease (LOD) (bottom row) by serotype (columns). Disease risk is relative to the null disease risk (NDR), defined as the risk among those with antibody concentrations below the lower limit of quantification, and scaled to 1. Curves show the maximum likelihood estimate; 95% confidence intervals (gray) are generated using the same maximum likelihood approach, except for step functions (serotype IV and V EOD and serotype II EOD and LOD), which use bounds from Donovan risk thresholds. Note: curves not generated for serotype IB and V LOD (< 5 cases above the lower limit of quantification).

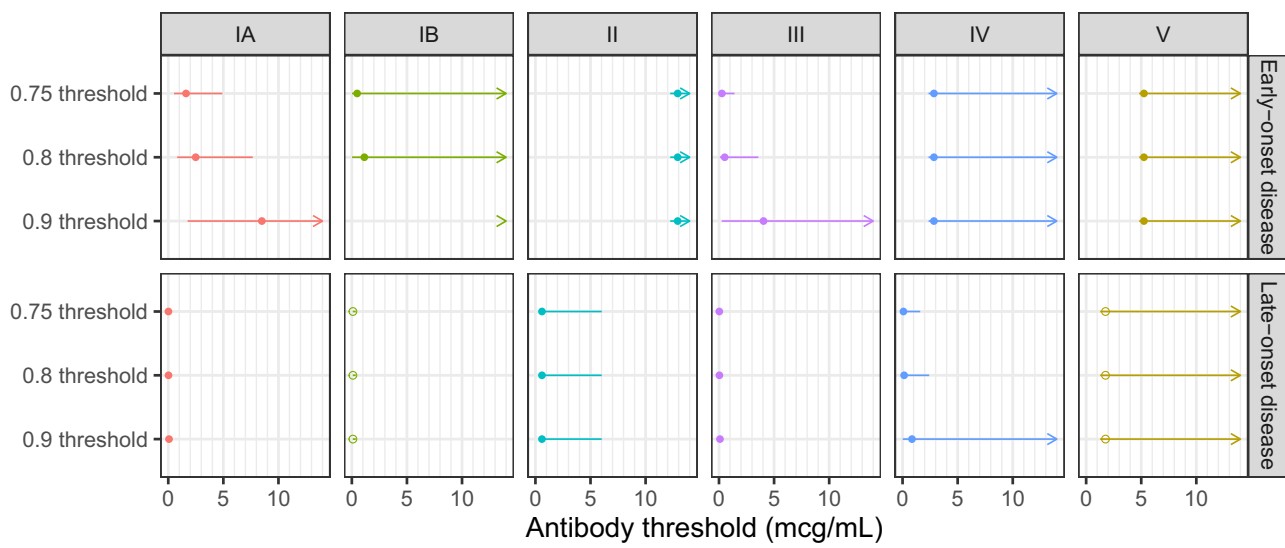

**Fig. 4 | Serotype-specific anti-capsular polysaccharide IgG protective thresholds (mcg/mL) and 95% confidence intervals for early-onset disease and late-onset disease.** 0.75, 0.80, and 0.90 protective thresholds (points) and 95% confidence intervals (lines) for each serotype (columns) for early-onset disease (EOD) (top row) and late-onset disease (LOD) (bottom row) derived from the risk curves generated with the covariate-adjusted logit model (CALM). Points show the maximum likelihood estimate. Lines indicate 95% confidence intervals, arrows indicate thresholds and/or upper bound beyond x axis. For step functions (serotype II, IV, and V EOD and serotype II LOD) and for those not meeting sample size criteria (serotype IB and V LOD), zero risk thresholds and bounds are used. Threshold estimates for those not meeting sample size criteria (serotype IB and V LOD) have open circles, as opposed to filled in.

Curves generated in sensitivity analyses assessing the timing of blood spot collection in relation to disease onset, the timing of antenatal screening, the role of IAP, alternate controls for LOD comparisons, gestational age, (Figs. 5–7 and Supplementary Fig. 7) and race/ethnicity (Supplementary Fig. 8) generally fell within the 95% bounds of the primary analysis curves, and in many cases were entirely overlapping with primary analysis curves (Figs. 5–7). In addition, curves generated with Dunning's scaled logit model, using the entire study sample (i.e., not restricted to those without missing key covariates), did not differ substantially from primary analytic curves (Fig. 7).

**Table 3 | Protective thresholds by serotype and age at onset**

| Serotype | Age at onset | 0.75 protective threshold (mcg/mL) | 0.8 protective threshold (mcg/mL) | 0.9 protective threshold (mcg/mL) | Zero risk protective threshold (mcg/mL) |
|---|---|---|---|---|---|
| IA | EOD | 1.61 (0.53–4.92) | 2.49 (0.81–7.67) | 8.49 (1.76–41.01) | 9.08 (8.51–11.78) |
| IA | LOD | 0.01 (0–0.08) | 0.02 (0–0.11) | 0.07 (0.02–0.29) | 1.25 (0.94–30.82) |
| IB | EOD | 0.48 (0.01–16.92) | 1.13 (0.02–70.83) | 16.34 (0.04–6052.37) | 6.54 (4.85–168.06) |
| IB | LOD | Sample size <5 above lower limit of quantification so curve not generated and zero risk protective threshold used | Sample size <5 above lower limit of quantification so curve not generated and zero risk protective threshold used | Sample size <5 above lower limit of quantification so curve not generated and zero risk protective threshold used | 0.10 (0.09–0.36) |
| II | EOD | 12.91 (12.24–43.37)* | 12.91 (12.24–43.37)* | 12.91 (12.24–43.37)* | 12.91 (12.24–43.37) |
| II | LOD | 0.59 (0.53–6.02)* | 0.59 (0.53–6.02)* | 0.59 (0.53–6.02)* | 0.59 (0.53–6.02) |
| III | EOD | 0.27 (0.05–1.41) | 0.52 (0.07–3.57) | 4.04 (0.26–64.04) | 1.53 (1.31–22.17) |
| III | LOD | 0.03 (0.01–0.06) | 0.04 (0.02–0.08) | 0.09 (0.05–0.17) | 0.92 (0.7–21.8) |
| IV | EOD | 2.85 (2.36–39.97)* | 2.85 (2.36–39.97)* | 2.85 (2.36–39.97)* | 2.85 (2.36–39.97) |
| IV | LOD | 0.08 (0–1.60) | 0.14 (0.01–2.41) | 0.85 (0.01–72.79) | 0.87 (0.66–21.28) |
| V | EOD | 5.25 (4.81–35.22)* | 5.25 (4.81–35.22)* | 55.25 (4.81–35.22)* | 5.25 (4.81–35.22) |
| V | LOD | Sample size <5 above lower limit of quantification so curve not generated and zero risk protective threshold used | Sample size <5 above lower limit of quantification so curve not generated and zero risk protective threshold used | Sample size <5 above lower limit of quantification so curve not generated and zero risk protective threshold used | 1.76 (1.27–50.6) |

*Step function so zero risk protective threshold used.

Curves excluding specimens collected before 2013 also did not differ from primary analytic curves (Fig. 5), supporting inclusion of all specimens in the primary analysis. Curves generated with isotonic regression were similar to CALM-generated curves, indicating that the logit function fit the data well (Fig. 7). Finally, protective thresholds generated using other statistical methods (the Bayesian absolute disease rate and weighted logistic regression methods) also found higher protective thresholds for EOD than LOD and for serotypes II, IV, and V compared to the other serotypes (Supplementary Fig. 9, Supplementary Table 3).

## Discussion

Our case-control study provides the strongest evidence to date that newborn anti-CPS IgG antibody concentrations can distinguish infant GBS cases from controls and be leveraged to describe the shape of the relationship between anti-CPS IgG antibody concentrations at birth and risk of infant GBS disease. This is a key step towards considering serological thresholds of risk reduction as potential endpoints for trials evaluating capsular polysaccharide-based maternal GBS vaccine candidates. We also provide the most robust data to date that protection thresholds vary by serotype and by age at onset, with notably higher antibody concentrations required to protect against EOD versus LOD.

Our study has several strengths. First, our study enrolled eight times more cases than currently published GBS seroepidemiologic studies using the standardized assay; this enabled us to generate risk curves and thresholds for a wide range of serotype and age at onset strata, including the first estimates specific to serotypes Ib, II, IV, and V. Second, the majority of blood spots were collected within the first 48 hours of life, minimizing concerns, particularly for LOD, about bias due to decay of maternally transferred antibodies or the infant's own immune response after disease onset. Third, risk curves and thresholds were adjusted for key covariates, a gold-standard practice in observational case-control analyses; while covariate adjustment did not have a strong influence on curve shape, it did improve curve fit[24]. Fourth, the sensitivity analyses we conducted to explore the potential impact of a wide range of concerns, from DBS storage time before processing to widespread IAP use in the US setting, suggested our results were robust to these factors. Finally, we applied multiple analytic methods for protection threshold estimation as a secondary method, and these yielded similar trends in results (Supplementary Fig. 9).

The low protective thresholds for LOD that we observed across serotypes are consistent with a previous study that reported LOD thresholds based on a much smaller sample size and a non-standardized assay[25]. Low LOD protective thresholds are biologically plausible given that vertically-transmitted EOD is often associated with exposure to a high bacterial load in utero and during passage through the birth canal, whereas common routes of LOD acquisition, such as skin-to-skin contact with caregivers, breastmilk colonization, or healthcare-associated transmission in a NICU setting, are likely associated with lower intensity bacterial exposures[26]. It has also been speculated that antibodies in breast milk may contribute to the protection of nursing infants, and may partially explain the low LOD thresholds we observed[27]. The observation that GBS LOD may be readily preventable by a minimal boost in antibody concentrations lends hope to the feasibility of maternal vaccination as a strategy to prevent other leading bacterial causes of late-onset neonatal sepsis, given that similar exposure dynamics may be involved.

Our observation that risk curves and thresholds differed across disease-onset and serotype strata suggests that a single universal threshold may not be appropriate, although this may be a practical solution for late-onset disease, or for a subset of serotypes. The variation observed underscores limitations to the interpretation of thresholds derived from aggregate data. Curves and thresholds derived from EOD and LOD cases combined likely underestimate antibody concentrations required to protect against EOD. Aggregate EOD and LOD thresholds also complicate comparisons of estimates across studies and geographical settings since the ratio of EOD to LOD can vary markedly, at least partially due to IAP implementation[28]. Similarly, serotype distribution varies by geography[28], and we found wide heterogeneity by serotype in EOD protective thresholds, with serotype II up to 60 times higher than the more common serotypes Ia and III. Additionally, anti-CPS IgG antibody concentrations against the minor serotypes IV and V were more similar between cases and controls than for other serotypes. Investigations of the underlying causes behind serotype differences in protective thresholds will provide important context, as ours is not the first study to observe variability in protective thresholds by serotype[29–31]. While we are not aware of other seroepidemiology studies that have characterized the full shape of the antibody concentration-disease risk relationship, anti-CPS IgG-based correlates of protection against invasive pneumococcal disease have been shown to vary notably by serotype,

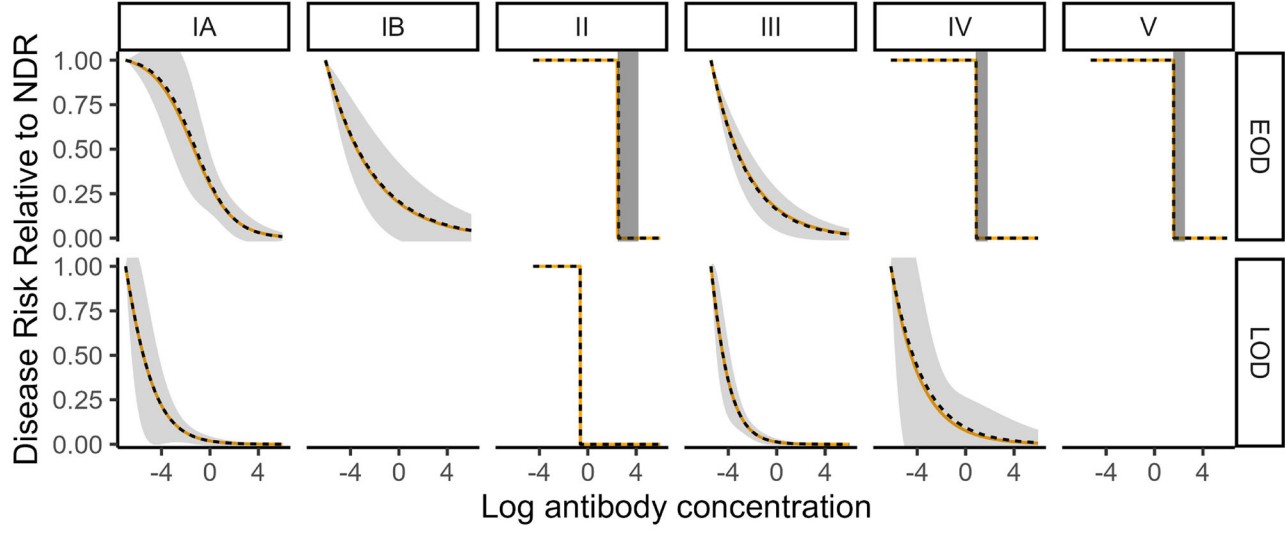

Sensitivity analysis — Blood spot timing — Primary

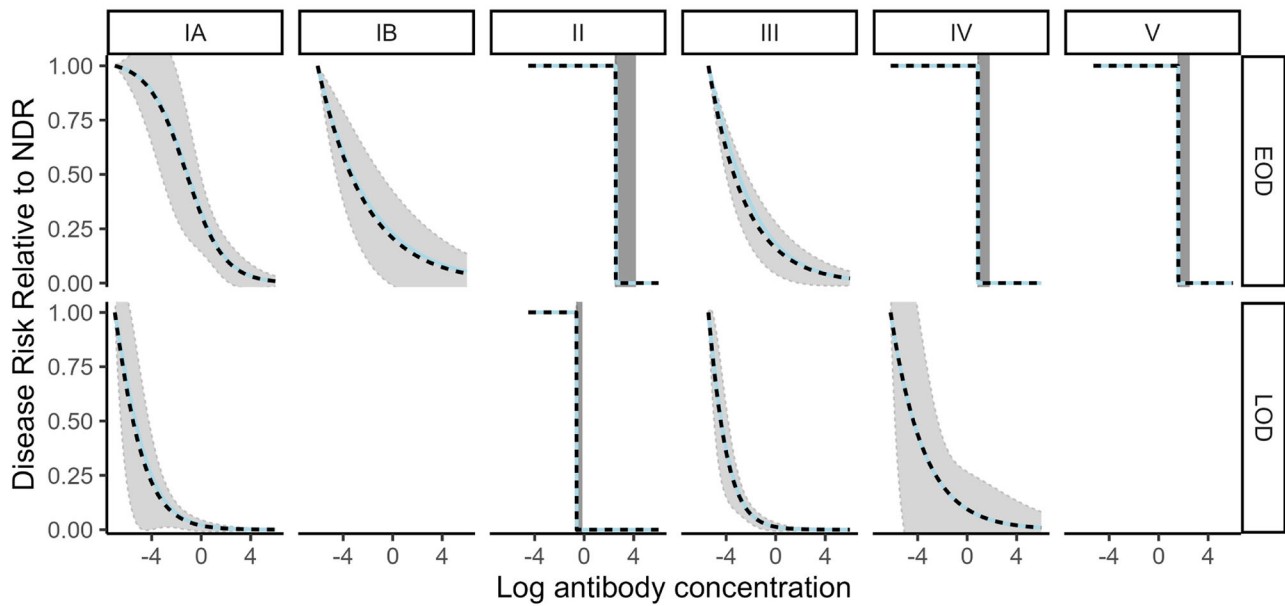

Sensitivity analysis — Pre 2013 — Primary

**Fig. 5 | Risk curves for anti-capsular polysaccharide IgG antibody concentrations from sensitivity analyses for timing of blood spot collection and duration of blood spot storage.** Colored lines show the following sensitivity analyses: timing of blood spot collection relative to disease onset (top panel): "Blood spot timing" curves (yellow) were generated using the CALM method after exclusion of $n = 27$ cases (23 EOD and 4 LOD) with samples collected more than 3 days post disease onset. Duration of blood spot storage (bottom panel): "Pre 2013" curves (light blue) were generated using the CALM method after exclusion of $n = 31$ case blood spot samples collected before 2013—(see supplementary information). Black dotted risk curves for the primary analysis were generated using the covariate-adjusted logit model (CALM), adjusted for gestational age, intraamniotic infection, and study site, and are shown for early-onset disease (EOD) (top row) and late-onset disease (LOD) (bottom row) by serotype (columns). Disease risk is relative to the null disease risk (NDR), defined as the risk among those with antibody concentrations below the lower limit of quantification, and scaled to 1. Curves show the maximum likelihood estimate; 95% confidence intervals (gray) are generated using the same maximum likelihood approach, except for step functions (serotype IV and V EOD and serotype II EOD and LOD), which use bounds from Donovan risk thresholds. Note: curves not generated for late-onset disease for serotypes IB and V (<5 cases above the lower limit of quantification).

suggesting variation across serotypes is not unusual[32]. Functional antibody data may provide additional insights, and we are exploring options for functional assay testing of our specimens, including modification of the Group B Streptococcus Assay standardization (GASTON) Consortium-endorsed opsonophagocytic killing assay for use on DBS[33].

The availability of the GASTON standardized, validated assay for anti-CPS IgG antibody concentration assessment opens the door for threshold comparisons across studies and potential for meta-analysis. Unfortunately, studies using the GASTON assay to date differ from each other and from our study in design (whether controls were from

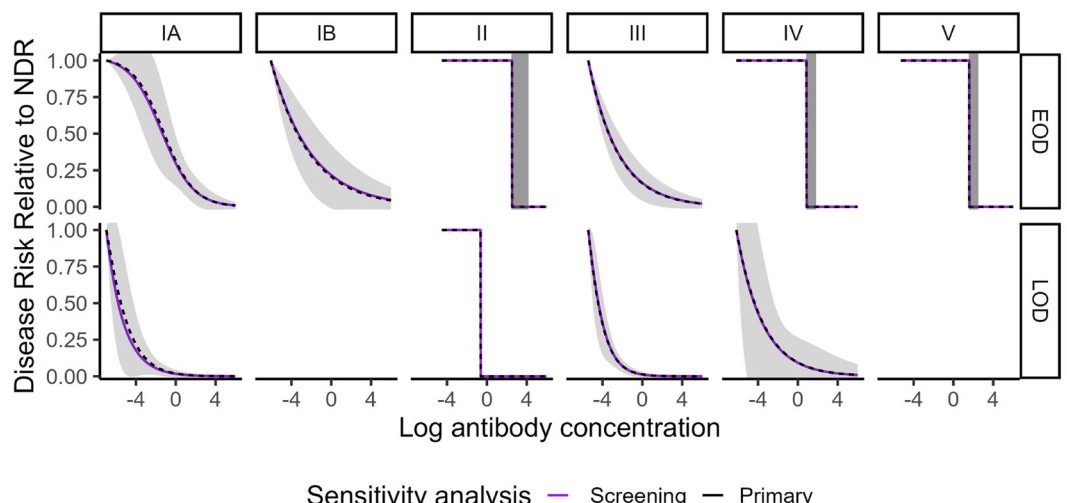

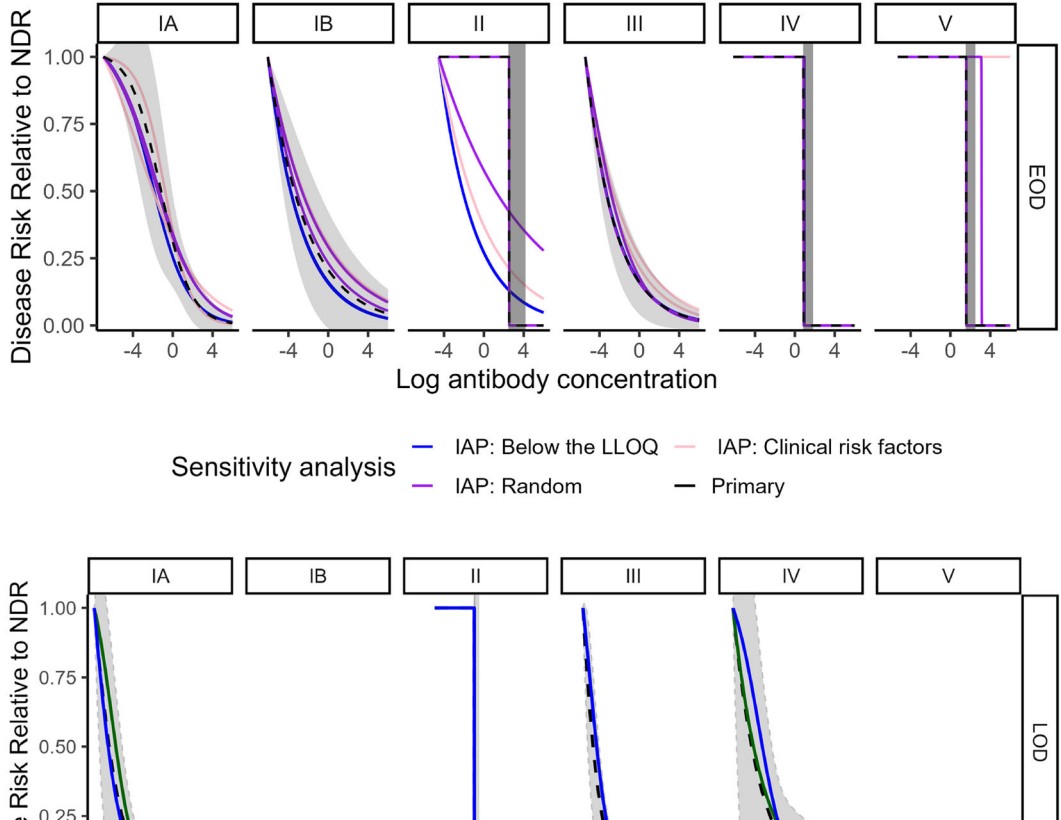

GBS colonized mothers or not), setting (U.S./high income/widespread IAP vs. South Africa/middle income/limited IAP vs. Finland/high income/limited IAP), specimen type (particularly whether specimens were collected at or near birth vs after disease onset), sample size to allow for key stratification by serotype and age at onset, and analytic methods for deriving risk curves and thresholds, including most notably whether the methods allowed for covariate adjustment[8,10].

Despite these differences, the thresholds estimated in these studies still fall within the range of the stratified thresholds estimated in our study. Ongoing studies will provide additional estimates and expand the possibilities for cross-study comparisons and meta-analyses[34,35].

Our study is subject to some limitations, largely attributable to efforts to maximize the study sample size. For example, to ensure a robust number of cases, we included cases from more birth years

**Fig. 6 | Risk curves for anti-capsular polysaccharide IgG antibody concentrations from sensitivity analyses for timing of antenatal screening, IAP, and alternate LOD control groups.** Colored lines show the following sensitivity analyses: Timing of antenatal screening (top panel): "Screening" curves (purple) were generated using the CALM method after exclusion of $n = 165$ controls screened more than five weeks prior to delivery. Intrapartum antibiotic prophylaxis (IAP) (middle panel): "Below the LLOQ" curves (blue) were generated after a small number of randomly selected controls who received 4 or more hours of IAP and had IgG concentrations below the LLOQ were treated as counterfactual cases (i.e., reclassified as cases for analytic purposes). "Clinical risk factors" curves (pink) were generated after a small number of randomly selected controls who received 4 or more hours of IAP and had clinical risk factors highly associated with EOD were treated as counterfactual cases. "Random" curves (purple) were generated after a small number of randomly selected controls who received 4 or more hours of IAP were treated as counterfactual cases. A maximum of five controls for each serotype were treated as counterfactual cases, with the exact number determined by

applying the rate of cases among colonized women in the absence of IAP: 11 per 1000 live births. All curves were generated using the CALM method and repeated twice to allow for variability in the random selections. Alternate LOD control group (bottom panel): "Sample 1" (green) and "Sample 2" (blue) curves were generated using an alternate control group designed to reflect the epidemiology of LOD—see methods section. Black dotted risk curves for the primary analysis were generated using the covariate-adjusted logit model (CALM), adjusted for gestational age, intraamniotic infection, and study site, and are shown for early-onset disease (EOD) (top row) and late-onset disease (LOD) (bottom row) by serotype (columns). Disease risk is relative to the null disease risk (NDR), defined as the risk among those with antibody concentrations below the lower limit of quantification, and scaled to 1. Curves show the maximum likelihood estimate; 95% confidence intervals (gray) are generated using the same maximum likelihood approach, except for step functions (serotype IV and V EOD and serotype II EOD and LOD), which use bounds from Donovan risk thresholds. Note: curves not generated for late-onset disease for serotypes IB and V < 5 cases above the lower limit of quantification).

(2010–2022) than controls (2018–2022), as long as DBS specimens were stored refrigerated or frozen. Although it was not feasible to conduct a 15-year formal longitudinal stability study, cross-sectional analyses of specimens with different storage times before processing revealed no consistent evidence of antibody decay over time (Supplementary Figs. 1–3), and a sensitivity analysis excluding pre-2013 cases detected no meaningful difference in risk curves or thresholds (Fig. 5; bottom panel). It is possible that our control antibody concentration distributions differed from those that might have been observed during the full study period if there were changes in maternal colonization patterns (e.g., serotype distribution) from 2010 to 2022. However, we are not aware of evidence of changing colonization patterns among US women, and stratifying analyses by serotype likely mitigated any impact on curves and protection thresholds. Controls also came from fewer states than cases due to enrollment challenges; however, distributions of antibody concentrations among controls by site were largely overlapping (Supplementary Fig. 10), and our models adjusted for study site. Due to widespread use of IAP, controls may also have included a small number of infants who would have been cases in the absence of IAP; when this was addressed in a sensitivity analysis of EOD, however, results remained similar. While DBS were not collected right at birth (i.e., not cord blood), more than 80% were collected in the first 48 hours of life, providing a strong approximation of antibody levels present at birth. Our use of DBS as the primary specimen also required adaptation of the standardized GASTON assay, but bridging studies demonstrated good correlation with serum value[36]. Further, while we enrolled the largest number of cases and controls of a GBS seroepidemiologic study to date, the sample size still limited the assessment of all strata of interest, such as LOD for serotypes Ib and V, which did not meet the sample size requirements to generate curves. Limited sample size already resulted in some instances in uncertainty around protective thresholds, and we note in particular the large uncertainty around the Ib EOD curve, which is likely driven by the small sample size of cases above the LLOQ and the presence of one outlier case with a high antibody concentration. Lastly, our study enrolled solely from the United States, and antibody concentrations among U.S. women may differ from those in other settings; even if antibody concentrations across populations vary, it may be that the relationships between antibody concentrations and disease risk are similar across settings, and this will be important to evaluate as seroepidemiologic data from different settings continue to accrue.

Our estimate of risk curves for anti-CPS IgG and protection thresholds expands the evidence available for decision-making regarding immunological endpoints for phase-III trials of capsular polysaccharide-based maternal GBS vaccine candidates[37]. If successful, maternal GBS vaccines would reduce illness and death due to neonatal sepsis, a persistent contributor to under-5 child mortality, and pave the way for other maternal vaccines to prevent bacterial infections in early

life[3,38]. They also provide a promising opportunity to address the inequitable disease burden suffered by infants in low-resource settings, as many countries in Africa and South Asia already have a platform for delivering maternal tetanus vaccines, and pregnant women are accustomed to vaccination during pregnancy.

## Methods
This study was reviewed and approved by CDC and site-level Institutional Review Boards (IRBs) (Supplementary Table 4) (See 45 Code of Federal Regulations (C.F.R.) part 46.114). Cases and controls were enrolled by individual-level parental consent when required, or through waivers of individual-level consent and de-identification when this alternate approach was approved (Fig. 1). Due to challenges with reaching parents to attempt individual-level consent, most cases, as well as controls from California and Georgia, were enrolled through waivers of consent. Participants did not receive any monetary compensation.

### Study population
Cases (from 2010–2022) and controls (from 2018–2022) were enrolled in eight ABCs population-based catchments in the United States: California (3 county Bay area), Colorado (5 county Denver area), New York (7 county Rochester area and 8 county Albany area), Portland, Oregon (3 county Portland area), and state-wide in Connecticut, Georgia, Maryland, and Minnesota[39]. Cases were defined as infants <90 days old with isolation of GBS from a normally sterile site; EOD cases had a first positive culture on days zero to six of life LOD cases on days 7 to 89 of life. Controls were defined as infants born to GBS-colonized women based on routine antenatal or intrapartum screening who did not develop invasive GBS disease in the first 90 days of life. Controls were identified at selected obstetric care facilities within surveillance catchments. To account for differences in colonization (controls) and invasive disease (cases) serotype distributions, we over-enrolled controls with the aim of a 1:3 case-to-control ratio for serotype-specific analyses.

To minimize potential misclassification, sites that used consented control enrollment confirmed the mothers' intent to reside in the catchment area during the three months after birth to enable capture of any EOD or LOD episodes through routine ABCs surveillance. Sites that enrolled de-identified controls were not able to confirm intent to reside in the catchment area, but they were able to identify any episodes of illness among controls. Through these procedures, we identified and excluded one enrolled control that subsequently developed invasive GBS disease.

### Study design
We used an unmatched case-control design stratified by GBS serotype and EOD vs LOD to characterize the relationship

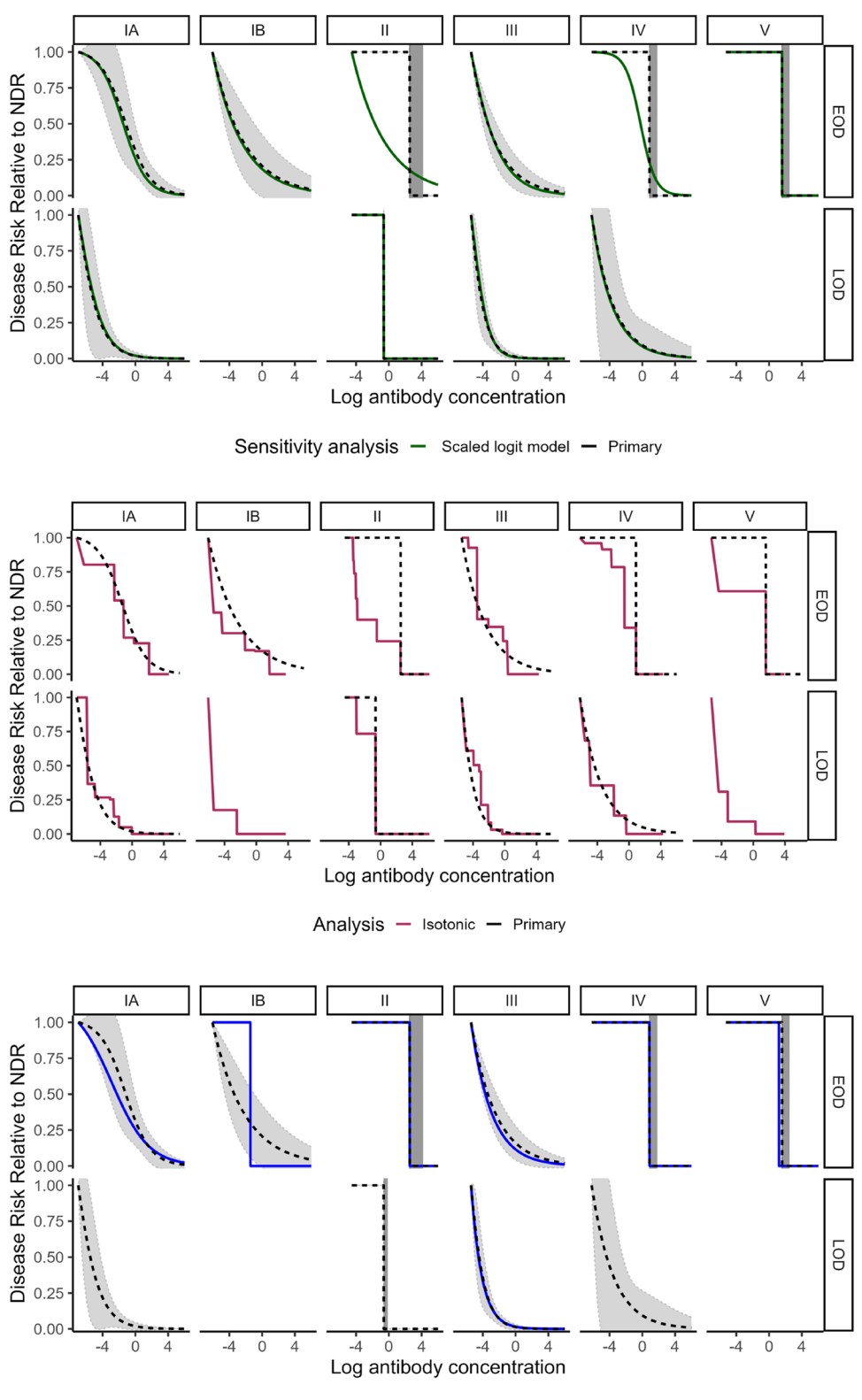

between newborn antibody concentrations close to birth (exposure) and infant GBS disease (outcome). Remnant DBS collected as part of routine newborn screening were used to measure antibody concentrations. The GBS serotype was characterized from the invasive disease-causing strain for cases and from remnant maternal screening specimens for controls, and is referred to as the target serotype throughout this report.

Case and control demographics and clinical characteristics were abstracted from the infant's medical chart and the mother's labor and delivery record using a standardized form. Maternal medical conditions, including underlying conditions routinely collected by ABCs, and intrapartum conditions (e.g., intraamniotic infection), were recorded as 'Yes' if noted in the hospital chart, 'No' when they were not noted, and 'Missing' if no chart was available (Table 1).

**Fig. 7 | Risk curves for anti-capsular polysaccharide IgG antibody concentrations from sensitivity analyses for missing data, curve shape, and gestational age.** Colored lines show the following sensitivity analyses: Exclusions due to missing data (top panel): "Scaled logit model" curves (green) were generated using scaled logit models and include all observations ($n = 268$ early-onset disease cases; $n = 375$ late-onset cases; $n = 2801$ controls). Curve generation method (middle panel): "Isotonic" curves (red) were generated using isotonic regression, a non-parametric approach that does not impose curve shape—see methods section. Gestational age (bottom panel): "34 weeks+" (blue) curves were generated after restriction to cases and controls with gestational age of 34 weeks or greater. Black dotted risk curves for the primary analysis were generated using the covariate-

adjusted logit model (CALM), adjusted for gestational age, intraamniotic infection, and study site, and are shown for early-onset disease (EOD) (top row) and late-onset disease (LOD) (bottom row) by serotype (columns). Disease risk is relative to the null disease risk (NDR), defined as the risk among those with antibody concentrations below the lower limit of quantification, and scaled to 1. Curves show the maximum likelihood estimate; 95% confidence intervals (gray) are generated using the same maximum likelihood approach, except for step functions (serotype IV and V EOD and serotype II EOD and LOD), which use bounds from Donovan risk thresholds. Note: curves not generated for late-onset disease for serotypes IB and V (< 5 cases above the lower limit of quantification).

## Specimens and laboratory methods

At CDC, GBS serotyping was performed on invasive disease-causing isolates for cases and on isolates from remnant maternal screening specimens for the majority of controls or directly on selective broth antenatal screening remnants for a subset of controls. Capsular serotype was ascertained by a range of equivalent methods: whole-genome sequencing and a validated GBS bioinformatics pipeline (cases from 2015–2022)[40]; latex agglutination and/or Lancefield precipitation tests using rabbit antisera to capsular polysaccharide types Ia, Ib, and II through IX with polymerase chain reaction (PCR) serotyping for non-typeable isolates (cases before 2015, initial control specimens)[41]; and real-time PCR on isolates or selective broth directly (majority of controls)[42].

State newborn screening laboratories provided remnant DBS specimens from routine newborn screening for both cases and controls. Systems were put in place to limit study DBS storage at room temperature to six weeks or less at newborn screening programs that did not routinely store DBS under refrigeration or freezing. Once received by CDC, DBS were stored at ≤−20 °C with desiccant until processing. Antibody concentrations in DBS specimens were measured using a previously described 6-plex (Ia, Ib, II, III, IV, and V) anti-CPS IgG direct Luminex immunoassay (dLIA) developed by Pfizer, Inc. (Pearl River, New York, USA), adopted by the international GASTON Consortium as their standardized assay[43,44], and modified for use on DBS eluent[36]. A bridging study was conducted to evaluate the use of DBS as a sample matrix for the established assay. The concordance correlation coefficient (CCC) was used to assess equivalence between $log_{10}$-transformed concentrations in paired serum and DBS eluents for a panel of 33 matched sera and DBS that spanned the dynamic range for each serotype-specific assay. CCC values were >0.99 with the lower bound of 95% confidence above 0.98 for each assay. To increase accuracy in DBS-to-serum equivalency, a serotype-specific conversion factor based on Deming regression results was applied if the 95% confidence interval (CI) on the intercept did not include zero (indicating that the baseline level between the two conditions had significant systematic differences) or if the 95% CI on the slope did not include one (indicating significant proportional differences between the two conditions). Conversion factors were applied for serotypes II, III, IV, and V[36].

Data were captured as median fluorescence intensities (MFI) using the Bio-Plex 200 system and converted to µg/mL antibody concentrations using a reference standard curve and accounting for the serum dilution factor. Assay results are reported in weight-based measurements of anti-CPS IgG antibodies (µg/mL)[45]. In preliminary natural immunity studies, a high proportion of specimens were found to fall below the lowest usable point of the standard curve or below the LLOQ. A comparison of antibody-depleted serum (ADS) MFIs values against assay blank MFI values established a ratio-based algorithm that, when applied to LLOQ samples, demonstrated that LLOQ samples performed similarly to the ADS samples tested and, thus, demonstrated sufficient assay sensitivity. Thus, for analytic purposes, specimens with antibody concentrations designated as below the LLOQ were assigned a value of half the serotype-specific LLOQ and are referred to as below the LLOQ specimens throughout this report.

We assessed the relationship between duration of DBS storage before testing and antibody concentration, since a meaningful portion (40%) of case specimens had longer storage times than controls, and stability of anti-CPS IgG analytes under long-term storage has not been demonstrated previously for the DBS matrix. We excluded target serotype antibody results (e.g., anti-Ia CPS IgG for those classified as Ia cases and controls exposed to maternal colonization with serotype Ia) from this analysis because differences in case and control antibody concentrations to the target serotypes were central to the study's primary objective analysis. We used scatterplots and LOESS smoothing to assess visually the relationship between antibody concentrations and storage duration. For each serotype and case/control stratum, we assessed two plots: one included all antibody concentrations in which a downward trend would be consistent with potential antibody degradation over time, and one excluded below the LLOQ observations to ensure that they did not obscure evidence of degradation among those with quantifiable concentrations. For all other analyses in this report, we used the target serotype only.

To assess whether antibody concentrations among DBS collected after disease onset differed from concentrations in specimens collected before disease onset, either due to an infant antibody response to disease or natural maternal antibody decay after birth, we analyzed the relationship between timing of infant specimen collection relative to disease onset and antibody concentration for EOD and LOD cases separately.

## Statistical analyses

**Descriptive analyses.** We generated a directed acyclic graph to identify potential confounders associated with both the exposure (i.e., infant anti-CPS IgG antibody concentration) and outcome (i.e., GBS disease) (Supplementary Fig. 11). Visual assessment of the relationship between potential confounders and antibody concentration among controls (Supplementary Fig. 10) was performed. Antibody concentration values were log-transformed to reduce the impact of outliers and approximate a normal distribution; log-transformed values were used for all primary analyses.

For each target serotype, we compared antibody concentration distributions between cases, stratified by EOD and LOD, and controls using boxplots and bee swarm plots. Controls were categorized only based on the serotype of the colonized mother, and thus the same set of controls was used for both EOD and LOD serotype-specific analyses. To assess the ability of antibody concentrations to distinguish cases from controls, we constructed receiver operating characteristic (ROC) curves for each target serotype and age at onset stratum, conducted Wilcoxon rank sum tests, and calculated the area under the curve (AUC) for each ROC plot.

**Risk curves and protective threshold generation.** We generated 'risk curves' plotting the relationship between anti-CPS IgG antibody concentration (x-axis) and relative disease risk reduction (y axis), for all strata of target serotype and age at onset combinations that had at least five cases with antibody concentrations above the LLOQ. The

CALM method[46] (see Supplementary Methods Appendix A and Supplementary Fig. 12 for more details), an extension of Dunning's Scaled Logit Model (SLM)[47], is equivalent to the SLM if no covariates are included, and was used to generate risk curves. The CALM method allows disease risk among specimens below the LLOQ, defined as the null disease risk (NDR), to vary by covariate values and generates relative risk curves for anti-CPS IgG; we scaled the curves so the NDR was 1 at the LLOQ. We used maximum likelihood to estimate CALM's parameters and tested a range of starting values to ensure the algorithm did not converge to local maxima. For strata with less than five cases above the LLOQ (i.e., limited data to inform curve shape), we reported a protective threshold based on the maximum observed antibody concentration among cases as described by Donovan, et. al[48].

Covariates were considered for inclusion in CALM models if they were either identified as potential confounders (Supplementary Fig. 11) or suggested by the US Food and Drug Administration (FDA)'s Center for Biologics Evaluation and Research (CBER) during informal review. Gestational age (< 37 vs ≥37 weeks' gestation) and intraamniotic infection (defined as notation in the labor and delivery record of intrapartum fever, suspected chorioamnionitis, or chorioamnionitis) were included as potential confounders. The study site was included based on FDA input and grouped based on sample sizes (GA, CA, other). Some potential confounders were not included because of incomplete data (first delivery) or because they were too rare (maternal HIV, prior infant with GBS). Race and ethnicity were not included because they likely operate through multiple pathways with more proximal intermediary confounders already included in the model (gestational age) (Supplementary Fig. 11); however, these were evaluated in a sensitivity analysis. Considering the limited number of covariates available for inclusion, for simplicity, we included all pre-specified variables in CALM models. Primary CALM analyses excluded records with missing values for intraamniotic infection or gestational age. Evidence of interactions between these key covariates and antibody concentrations was also assessed.

Population disease rates used in CALM models to account for the case-control design of our study (disease rates were greater in our study sample, which largely consisted of infants of colonized mothers, than in the broader study population from where the cases arise) were assumed to be 0.88 EOD cases per 1000 live births and 0.58 LOD cases per 1000 live births (see Supplementary Methods Appendix B for more details).

Putative protective thresholds were calculated by identifying the first point on the curve at which the relative reduction was equal to or less than the pre-specified risk reduction (i.e., 75%, 80%, 90%). Standard errors of risk curves and corresponding protective thresholds were obtained through a first-order Taylor expansion of the curve as a function of the parameters and the estimated Hessian matrix[49]. Standard errors were used to calculate 95% confidence bounds of estimated curves and protective thresholds. If curves converged to a step function, we used the Donovan method for estimation of the zero-risk protective threshold and confidence bounds[48].

**Sensitivity analyses.** Sensitivity analyses were conducted using the CALM method as described above on specific subsets of the data outlined below, unless otherwise specified.

**Timing of blood spot collection relative to disease onset.** We conducted a sensitivity analysis of risk curves for anti-CPS IgG and associated protective thresholds that was restricted to cases with DBS collected within 3 days of disease onset, as has been done previously[10]. This analysis primarily pertains to EOD, as DBS were generally collected shortly after birth and well in advance of LOD onset. Results are

provided for LOD since DBS were collected >3 days after disease onset for a small number of LOD cases.

**Duration of blood spot storage.** We conducted a sensitivity analysis of curves and protective thresholds that excluded specimens collected before 2013 based on results of a cross-sectional analysis of specimen stability over time.

**Timing of antenatal screening.** We restricted controls to those whose mothers were screened within five weeks before delivery, given the possibility that those screened earlier may no longer have been colonized at birth, and thus their infants may have had no GBS exposure.

**Intrapartum antibiotic prophylaxis.** Because this study limited controls to infants born to women with positive antenatal GBS screening tests, most mothers of controls received adequate IAP (4+ hours of a beta-lactam). It is therefore likely that some controls would have been EOD cases in the absence of IAP. To assess how different risk curves and thresholds might be in a hypothetical setting without IAP, we conducted a sensitivity analysis. In this scenario, we treated two to five controls who received IAP per serotype as EOD cases (i.e., counterfactual cases) when generating curves and protective thresholds. The exact number of counterfactual cases was based on the expected rate of infant disease among colonized women in the absence of IAP: 11/1000[50]. The counterfactual cases were randomly drawn from subgroups of controls who received 4+ hours of IAP and had epidemiologically relevant characteristics: 1) those with antibody concentrations below the LLOQ (a group considered in the literature to be at highest risk for EOD) and 2) those with clinical risk factors strongly associated with EOD (i.e., intrapartum fever and/or intraamniotic infection), as well as 3) randomly selected controls. We repeated this process twice to allow for variability in the random selections. We also repeated this process to include controls who received 2+ hours of IAP in the pool of potential counterfactual cases since 2+ hours of IAP confers partial protection[51].

**Alternate LOD control groups.** We conducted a sensitivity analysis using alternate control groups designed to reflect epidemiologic observations that approximately half of LOD cases are born to mothers not colonized with the disease-causing GBS strain[52,53]. For this sensitivity analysis, we leveraged the multiplex nature of the antibody binding assay, which simultaneously generates results for all 6 serotypes (i.e., one target and five non-target results). The alternate LOD control groups were comprised of infants born to mothers colonized by the target serotype (50%) and infants whose mothers were not colonized by the target serotype (50%), with a 1:6 case-to-control ratio for serotypes Ia, II, and IV, and 1:2 for serotype III, similar to the ratios in the primary analysis. For serotypes Ia, II, and IV, these control groups were created by randomly selecting three target serotype-matched controls per case and three additional non-target controls per case, with the random selection weighted according to the control population serotype distribution. For serotype III, we used a random sample of 50% of the serotype III controls and sampled the same number from the non-serotype III controls. We repeated this process twice to allow for variability in the random selections.

**Exclusions due to missing data.** To assess the impact of excluding individuals with missing values for key covariates in our primary CALM analysis, we compared our primary CALM curves to unadjusted curves using the SLM, which allowed us to include all cases and controls regardless of missing values in key covariates.

**Curve generation methods.** To assess whether the logit shape assumed by CALM fits the data, we generated curves using isotonic regression, a non-parametric approach that does not impose curve

shape beyond the requirement that the curve is a monotonic function. Finally, we compared curves and protective thresholds generated with CALM to those generated with the Bayesian absolute disease rate and weighted logistic regression methods recently published[10]. Since the methods are not expected to yield identical estimates, particularly as CALM generates a relative curve and the other methods generate an absolute curve, the objective was to assess whether trends in curve shape and protective thresholds across serotypes and EOD vs LOD were similar across methods.

**Gestational age sensitivity analysis.** We conducted a sensitivity analysis restricted to cases and controls with gestational age of 34 weeks and greater in recognition of the potential for residual confounding in the primary CALM analyses, where strata-specific sample sizes limited gestational age categorization to <37 vs ≥37 weeks.

Statistical analyses were conducted in SAS and R (see Supplementary Code).

### Reporting summary
Further information on research design is available in the Nature Portfolio Reporting Summary linked to this article.

## Data availability
The data that support the findings of this study are not openly available to protect the privacy of the study subjects. Study data can be made available upon request, including completion of the Data Use Agreement. Data requestors can contact the corresponding author directly or submit a request here: Isolate and Data Requests | ABCs | CDC.

## Code availability
Code is available as a supplementary file.

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

## Acknowledgements

We are grateful to the following individuals for their invaluable contributions to this project: from the Centers for Disease Control and Prevention: Wei Xing, Jarad Schiffer, Melissa Arvay, Jasmine Varghese, Peter Browning, Elise Gowen, and Paula Rios; from the California Emerging Infections Program: Mirasol Apostol, Gretchen Rothrock, Arthur Reingold, and Alison Ryan; from the Colorado Department of Public Health and Environment Newborn Screening Laboratory: Abena Watson-Siriboe and Kristin Viart; from the Connecticut Department of Public Health: Susan Petit; from the Yale School of Public Health: Amber Maslar and James Meek; from the Georgia Emerging Infections Program, Emory University School of Medicine: Amy Tunali, Ingrid Zambrano, Erica Hazra, and Molly McAlvany; from the Georgia Department of Public Health: Melissa Tobin-D'Angelo; from the Department of International Health, Johns Hopkins Bloomberg School of Public Health: Terresa Carter; and from the Maryland Department of Health: Prakash Paudyal, Suhita Gayen nee' Betal, Ikenna Osuagwu, and Hanah Kim; from the New York State Department of Health: Rebecca Hoelzl, Michele Caggana, Alison Muse, Daniele-Marisa Stansfield, Adam Rowe, Suzanne McGuire, Bridget Anderson, Kari Burzlaff, Nancy Spina, Norma Tavakoli; and from the University of Rochester Medical Center: Sarah Caveglia and Dwight Hardy. We also recognize the technical support of the GASTON consortium, especially Kirsty Le Doare and Tom Hall. The findings and conclusions in this report are those of the authors and do not necessarily represent the official position of the Centers for Disease Control and Prevention. This work was supported in part through a grant awarded to the CDC Foundation from the Gates Foundation (INV-009090). The funders played no role in the study design, data collection, analysis, and interpretation of data, or the writing of this manuscript.

## Author contributions

All authors made substantial contributions to the conception or design of the work (J.C.R., R.K., S.B., N.S., Y.C., M.M.F., A.N.B., A.R.T., S.T., P.M., S.J.S.); or data acquisition (J.C.R., S.B., Y.C., M.M.F., A.N.B., A.E.M., S.T., J.N., L.B.A., J.S., K.R.V., M.B., K.E., P.F., A.P.H., L.H.H., L.J., J.L.N., S.T.O., C.O.C., J.V.R., S.A.S., A.R.T., H.H.N.W., L.M., F.A., B.A., Y.L., P.Y.P., J.R., J.E.S., T.T., P.M., S.J.S.), or data analysis (J.C.R., R.K., S.B., N.S., Y.C., L.M., F.A., L.T.J., Y.L., P.M., S.J.S.), or data interpretation (J.C.R., R.K., S.B., N.S., Y.C., L.H.H., S.A.V., L.T.J., S.J.S.). All authors participated in drafting the manuscript or reviewed it critically for important intellectual content. Each co-author has given final approval of the version to be published and agreed to be accountable for all aspects of the work.

## Competing interests

Dr. Harrison has served on scientific advisory boards and/or given lectures for Sanofi Pasteur, Pfizer, and GSK, and has served on a data and safety monitoring board for Merck. He receives no compensation other than reimbursement for any travel expenses. The remaining authors declare no competing interests.

## Additional information

[1]Division of Bacterial Diseases, National Center for Immunization and Respiratory Diseases, Centers for Disease Control and Prevention, Atlanta, GA, USA. [2]Epidemic Intelligence Service, Centers for Disease Control and Prevention, Atlanta, GA, USA. [3]Georgia Emerging Infections Program, Emory University School of Medicine, Atlanta, GA, USA. [4]Georgia Department of Public Health, Atlanta, GA, USA. [5]California Emerging Infections Program, Oakland, CA, USA. [6]Division of Research, Kaiser Permanente Northern California, Pleasanton, CA, USA. [7]The Permanente Medical Group, San Leandro, California, CA, USA. [8]Colorado Department of Public Health and Environment, Denver, CO, USA. [9]New York State Department of Health, Albany, NY, USA. [10]University of Minnesota Medical School, Minneapolis, MN, USA. [11]Yale School of Public Health, Connecticut Emerging Infections Program, New Haven, CT, USA. [12]Department of International Health, Johns Hopkins Bloomberg School of Public Health, Baltimore, MD, USA. [13]Department of Pediatrics, University of Colorado School of Medicine, Denver, CO, USA. [14]University of Rochester Medical Center, Rochester, NY, USA. [15]Oregon Health Authority, Portland, OR, USA. [16]Connecticut Department of Public Health, Hartford, CT, USA. [17]Seneca Federal Health, Chantilly, VA, USA. [18]IHRC, Inc, Atlanta, GA, USA. [19]These authors contributed equally: Julia C. Rhodes, Rebecca Kahn. ✉e-mail: icq0@cdc.gov; stephanieschrag@outlook.com

