## [Transparent Peer Review file · Nature Communications]

A US case-control study to estimate infant group B streptococcal disease serological thresholds of risk-reduction

Corresponding Author: Dr Rebecca Kahn

Version 0:

Reviewer comments:

Reviewer #2

(Remarks to the Author)

The authors have adequately responded to the majority of the previous Reviewers comments and made the necessary edits in the manuscript. Also, the additional analysis undertaken, including in relation to LOD is appreciated. Remaining issues that warrant further consideration are:

1. Reviewer 2 , Comment 2 response: Whilst agreeing that IAP is not a confounder, as would not affect antibody concentration, there still is an inherent bias to only have included controls with titres >LLQ in the sensitivity analysis. As evident from the study, even among cases, there is a wide dispersion of IgG, including some cases with high titres. Similarly, at the population level- one would expect a wide dispersion of titers among controls, and it biases toward a higher STRR by only including controls with detectable titres. I would request that the sensitivity analysis include a random selection of controls, agnostic to antibody titres.

2. Re authors response to Reviewer 2 , Comment 2 – it is correct that there are risk factors for invasive GBS, with maternal GBS colonization being the major risk factor for EOD. However, a fair proportion of cases do still occur randomly, as risk factors are not identified in all cases even in the presence of IAP. If that was not the case, the rationale for selection of controls as done in studies such as this would be questionable- as controls will be inherently at no/low risk of EOD to follow through with the argument of cases not occurring randomly. Also, whilst its correct as shown in the current and earlier studies, that serotype specific IgG is inversely related to risk of invasive, this does not detract from the population heterogeneity (more so in controls) of antibody titres in case and controls. As such, disagree that controls should be only chose from those with measurable antibody titres, as opposed to random selection.

3. Rev 2 Comment 8 (and Rev 3 comment 3)- Both reviewers highlight the difference in time-period over which cases and controls were enrolled. Whilst the authors agree that there could be temporal changes in colonization, the effect thereof on temporal changes in antibody concentrations is not considered. As maternal colonization induces antibody responses, the temporal changes in serotype-specific colonization would also be associated with temporal changes in antibody titers. Consequently, the potential effect thereof warrants discussion, as its beyond the study framework to be able to account for such (other than another sensitivity analysis limiting controls to being temporally linked to cases).

(Remarks on code availability)

Reviewer #4

(Remarks to the Author)

The authors performed a large sero-epidemiologic study of GBS disease across eight sites in the United States. The authors analysed the data with a variety of appropriate statistical methods. Their manuscript was rigorously reviewed by several

reviewers, and the authors responded to all of the reviewers' suggestions for improvements, making the majority of the suggested changes and clarifications, and respectfully defending their position for the remainder. The review process has further improved the overall analyses and narrative of this study. This report of their findings is likely to prove highly valuable in the ongoing assessment and future implementation of GBS vaccines.

(Remarks on code availability)

Version 1:

Reviewer comments:

Reviewer #2

(Remarks to the Author)

Thanks for addressing the comments

(Remarks on code availability)

Outside of my expertise to review the coding

Reviewer #1 (Remarks to the Author):

This study uses a case control design to characterize the relationship between serotype specific anti-capsular polysaccharide (CPS) IgG concentrations and infant GBS disease risk. The findings can inform Phase III maternal GBS vaccine immunogenicity endpoints. **The study is difficult to interpret in its current state, due to highly statistical language and a lack of discussion on the part of the authors about what their results may mean in the real world.**

Response: Thank you for this feedback. Based on your helpful suggestions below we have simplified the language throughout, clarified the objective of the study, and discussed the real world implications—in short, the ‘real world’ implications of the study are that it informs maternal vaccine trial design and regulatory decision-making by providing needed information on the shape of the association between antiCPS IgG at birth and early or late onset disease risk. In fact preliminary data from the study have already been requested by regulatory authorities and industry sponsors for just this purpose. We also believe some of the variation by serotype that we documented may motivate research into mechanistic or causal factors that underlie this variation.

Introduction:

Comment 1: The summary of GBS epidemiology and the effectiveness of IAP is thorough. **The authors may want to consider adding information on what proportion of cases of EOD and LOD die.**

Response: We have added information on case fatality rates for EOD and LOD in the first paragraph of the introduction – see lines 74-77.

Methods

Comment 2: Would it be possible for the authors to conduct a sensitivity analysis to assess the potential impact of misclassification? Perhaps using data from Georgia, where re-identifying cases was possible, to extrapolate to the rest of the study?

Response: To clarify, the potential misclassification was only for the controls from California—Georgia was able to reclassify if needed but no cases among controls were identified; the sites that used consented control enrollment (CO, CT, MD, MN, NY) confirmed the mothers’ intent to reside in the catchment area during the three months after birth to enable reclassification as cases after capture of any EOD or LOD episodes through routine ABCs surveillance; no such instances were found. Because the CA site accounted for 39% (1081/2802) of 2802 controls, we agree with the reviewer’s concern.

Fortunately, since our manuscript submission this site was able to re-examine control medical records to identify EOD or LOD episodes and identified one control who became a case.

The curves and thresholds in the current re-submission were generated after exclusion of this control. Given these additional efforts, we are confident that misclassification of cases as controls did not impact study findings.

Comment 3: I assume that the CCC correlated the concentration of antibody, but it would be helpful to be more direct about this in the text, seeing as the reference is not yet published. Also – does the ≥ 0.98 have a confidence interval that can be included, or another measure to understand how results varied?

Response: The manuscript describing the correlation between serum and DBS is available on a preprint server (reference now included in the paper) and we hope to have an in press citation soon. We added text in the Methods to provide more detail about the CCC – see lines 177 tracked changes version; 172 – clean version.

Comment 4: Also related to the CCC: was the panel of 33 matched sera for all 6 serotypes? This would be a bit concerning because it would be ~6 DBS for each, which is a very small number. It would be good to add some info in this section in the absence of a published methods manuscript.

Response: As per above the detailed manuscript is available on a preprint server and cited. We added detail to the Methods about the panel - the full panel was used for each serotype and spanned the dynamic range for each serotype (see lines 177 tracked changes version; 172 – clean version).

Comment 5: Can the authors comment on the reasons for the exclusion of risk factors like PROM or previous baby with GBS infection in the list of confounders for the CALM models?

Response: Supplemental Figure 1 provides a directed acyclic graph that outlines our assessment of the relationship of the covariates to the primary exposure (anti-CPS IgG at birth) and the primary outcome (infant EOD or LOD). Being born to a mother who had a previous infant with GBS disease in this DAG was considered a possible confounder, but as stated in the methods (line 264 – tracked changes version ; line 253 – clean version), this characteristic was exceedingly rare in the study population and thus could not be included in CALM models. PROM along with IAP, duration of membrane rupture, and delivery mode, were all considered to affect an infant's risk of

GBS disease but not to be associated with infant antibody level at birth, and thus not to meet the criteria to function as confounders.

Comment 6: It would be helpful to add information in the methods about which statistical methods were used for the sensitivity analyses.

Response: We added a description of the sensitivity analysis methods at the start of the sensitivity analysis section in the Methods – see line 287 – tracked changes version; 276 – clean version. We also include a supplemental methods section which provides more details on the CALM method.

Results

Comment 7: Can the authors further clarify line 350? It's not clear what they considered a sufficient sample size and how this was determined.

Response: Thank you we have clarified the wording – see line 401 – tracked changes version; line 386 – clean version.

Comment 8: The paragraph describing figures 3 and 4 needs to be written a bit more clearly – while it is clear from a statistical perspective, it is hard to interpret/understand without more information. For example, step functions or “models converged to a clear solution” are not commonly used terms in medicine or ID. Similarly, the figure legends for Figs 3 and 4 need a bit more of a description, beyond the variables included and model used, and NDR needs to be defined in the vertical axis title or legend for fig 3.

Response: We revised this paragraph to use plainer language that would be understandable to non-statistician readers – see lines 401-420 in tracked changes version; 386-401 in clean version. We have added additional details to the legends, updated the y-axis labels to say, “Disease Risk Relative to NDR”, and defined NDR in the legends.

Discussion

Comment 9: Based on the data from the results, I am having trouble agreeing with the statement that this study presents strong evidence that anti-CPS IgG can serve as an immunological endpoint for trials.

Response: The objective of our study was to evaluate whether infant GBS cases (stratified by EOD and LOD and serotype) can be distinguished from controls based on anti-CPS IgG concentration, and if so to characterize the relationship between antibody

concentrations and disease risk to allow for determination of serological thresholds of risk reduction. While this objective is central to defining potential immunologic endpoints, other factors may also be relevant to endpoint decision-making (for example, elucidation of the causal mechanism by which anti-CPS IgG confers protection, the mechanisms underlying variation across serotypes, whether factors other than anti-CPS IgG confer protection, and whether vaccine-induced antibodies behave similarly to natural antibodies). We understand and sympathize with the reviewer's interests to understand the causal mechanisms underlying our observations; however, exploration of these was beyond the scope of our study and discussion of these would be pure speculation. We revised the opening of the discussion to focus on our study's direct findings and not to extend beyond to immunologic endpoint determinations.

Comment 10: I think the authors need to do a lot more interpretation in their Discussion as to what they are seeing in Figures 3 and 4. What is the significance/interpretation of curves (ie – 1a EOD) vs steps (II EOD, IV EOD, V EOD, II LOD) in figure 3? What do these suggest about how antibody correlation to protection might work? For Figure 4, I observe huge ranges of protective thresholds, which again, make interpretation difficult. For figure 4, it would be helpful to add a 0.5 and 0.25 threshold to the figures.

Response: We have added more interpretation to the fourth paragraph of the discussion, both to the start and the latter portion of that paragraph to provide additional interpretation. Briefly we believe the variation across serotypes and in curve shapes is not unexpected (variation across pneumococcal serotypes in anti-CPS IgG -based correlates of protection has been well demonstrated, for example Andrews et al 2014 cited in the MS references). Rather, we think the variation we documented suggests caution in aggregating across serotypes or disease onset strata.

Regarding inclusion of 0.5 and 0.25 thresholds in figure 4: we can provide these if the journal requests as our curves allow for threshold generation at any level. That said, we do not see immediate utility in including those—we currently include the three thresholds regulators have directly expressed interest in, and it is highly unlikely that the lower thresholds the reviewer proposes would be used to inform immunologic endpoints.

Comment 11: The authors discuss the fact that other studies have also found heterogenous results by serotype. It's very difficult to conclusively rule out low sensitivity of laboratory methods as a reason for this – unless there is a biological reason related to the make up of the capsule for different serotypes. I think the authors should expand on potential reasons for the heterogeneity that they observe.

Response: Again as stated above, and documented in the Andrews 2014 pneumococcal paper that derives correlates of protection directly from disease endpoint trial data, variation across serotypes in anti-GBS CPS IgG based thresholds has been seen for pneumococcus and we do not agree with this reviewer that the variation we see for GBS is a surprise or anomalous finding. Additionally, as to the question regarding whether low sensitivity of the antibody binding assay could have generated what may be a spurious finding, we describe in the methods that characterization of antibody-depleted serum (ADS) demonstrated that samples with low antibody concentrations performed similarly to the ADS samples tested (see line 195 in tracked changes version; 187 in clean version). A report describing this work in detail will be available on a preprint server in time to be cited in the current manuscript. Below are some additional details about our work with ADS for editor consideration.

A comparison of ADS MFI values against assay blank MFI values established a ratio-based algorithm that, when applied to samples flagged as LLOQ, identified two sets of samples: A) ones that would remain below the LLOQ regardless of repeated testing; B) ones that needed additional data analysis. Before the above algorithm was applied to 17,538 samples with naturally occurring antibody concentrations (i.e., real world samples), the percentage of unresolved LLOQ samples was ~47% (see Table 1). However, after application of the algorithm, the percentage of unresolved LLOQ samples dropped to 0.07%, thus demonstrating that the MFIs of ADS samples were similar to samples previously flagged as below LLOQ. This shows that if flagged by the assay as under the LLOQ, there's a high likelihood (99.93%) that the antibodies indeed are at unquantifiable levels similar to those observed in antibody-depleted serum.

Table 1. Results of ADS-applied ratios to natural immunity studies results

Study	Study sample size (N)	Total number of data calls (nx6 serotypes)*	# of unresolved LLOQ samples before application of ADS-derived ratios	# of Unresolved LLOQ samples after application of ADS-derived ratios
CHOP seroepi study**	2406	14,436	5,904 (40.9%)	4 (0.01%)
Dried blood spot study***	3837	23,022	11,634 (50.53%)	22 (0.09%)
Total	6243	37,458	17,538 (46.82%)	26 (0.07%)

*Each sample was tested for 6 analytes/serotypes

**oral presentation: Flannery D. Group B *Streptococcus* antibodies and placental antibody transfer ratios among persons presenting for childbirth 2019-2021. Pediatric Academic Societies 2023. April 28, 2023. Washington DC.

***Study described in this manuscript

Comment 12: In the list of limitations, the authors should consider the possibility that antibody concentrations of Americans may be quite different from those in developing settings, who are known to have a lower colonization prevalence, possibly to more frequent exposure and antibody development as a result.

Response: We agree with this point and have added some comment to this effect at the end of the limitations paragraph in the Discussion.

Reviewer #2 (Remarks to the Author):

The manuscript reports on the association of Group B streptococcus serotype specific anti-capsular antibody and risk reduction of invasive GBS disease. In an unmatched case-control study, leveraging off the CDC ABC surveillance system of invasive bacterial disease and newborn dry blood spots samples obtained as part of newborn screening for diseases, the authors provide important data on serotype-specific IgG predictive of various percentage risk reduction of invasive GBS disease. The study corroborates previous observations on the inverse association between serotype-specific IgG and invasive GBS disease, including differences in thresholds for early-onset and late-onset disease. The novelties of the study include, largely due to being the largest study of its kind in number of enrolled cases, having been able to report thresholds associated with risk reduction for each of the six serotypes included in a GBS polysaccharide-protein conjugate vaccine which is progressing to phase III. The data will likely inform regulatory considerations on having such a vaccine licensed based on serological markers (and safety).

Response: Thank you, this captures the study well and the unpublished data have already been of great interest to regulators in the U.S., Europe, UK and Canada

General comments:

Comment 1: Although traditionally studies of this nature have used the term “correlates of protection”, strictly speaking what is being studied here and elsewhere are serological correlates associated with risk reduction (SCRR) of invasive disease. This distinction is important, as true correlates of protection for a vaccine can only be determined from vaccine efficacy trials and could very well differ compared with sero-epidemiology studies (as was demonstrated for Haemophilus influenzae type b as an example). It is recommended that the term “correlate of protection be changed to “serological threshold of risk reduction” in the title and throughout the manuscript. The authors on occasion already use the terms interchangeably in the manuscript.

Response: We appreciated this comment and throughout the text have replaced the term “correlates of protection” with the term “serological thresholds of risk reduction” in line with this reviewer’s suggestion. We defined ‘protective threshold’ as the first point on the curve at which the relative risk was equal to or less than the pre-specified relative risk reductions of 75%, 80%, and 90% and use this term afterwards in several instances for simplicity - see line 278 in track changes version; line 267 in clean version) .

Comment 2: The authors correctly articulate a number of limitations/challenges to the study, and have undertaken a series of sensitivity analysis to address the limitations. Perhaps one of the key limitations is the standard of care of screening for GBS colonization in later 3rd trimester in the US, coupled with the use of intrapartum antibiotic prophylaxis (IAP), which has been demonstrated to reduce the risk of early onset disease by 90%. The ubiquitous use of IAP, innately results in the risk of EOD being substantially reduced. The authors have used a cut-off of IAP ≥ 4 hours before birth as being “adequate”, although there is a suggestion that IAP even within 2 hours of birth could be effective (which should be considered for an additional sensitivity analysis).

Response: While 2+h of IAP is not considered ‘adequate’ in US prevention guidelines (approximately 47% vs 90% effectiveness for 4+hour of beta lactam IAP), we agree with the reviewer that it does confer some protection and now additionally include a sensitivity analysis in the supplement looking at this as suggested. This analysis showed similar results to our original IAP sensitivity analysis, with curves generally falling within the 95% bounds of the primary analysis curves, and in many cases entirely overlapping with primary analysis curves.

Comment 2 con’t: The authors address this major confounder through a sensitivity analysis, whereby controls who have IgG below the limit of quantification are modelled to be cases for the relevant serotype. The concern, however, is that the distribution of IgG in the confirmed cases demonstrate heterogeneity, with a modest to high percentage for each of the serotypes being $>LLQ$. Consequently, the sensitivity analysis would be better suited through the random selection of controls irrespective of the IgG levels, or at the least to somewhat mimic what was observed for the cases. Given the what a major confounder IAP I in relation to EOD, one might well argue that should there be a difference between the current and proposed sensitivity analysis, the sensitivity analysis should be promoted as the appropriate measure for SCRR.

Response: We agree with the reviewer that IAP is a potential covariate of concern but don’t agree with these suggestions, although we would be willing to do these if the journal feels strongly. We outline our rationale here:

- In our DAG (supplemental Figure 1) we document that we don’t view IAP as a confounder (the reviewer states ‘major confounder’)—this is because IAP definitely affects risk of infant disease (greatly reduces it) but it does not to our understanding affect infant antibody concentration at birth, and a confounder needs to affect both the exposure and the outcome.

- We designed our sensitivity analysis for the counterfactual world with no IAP based on available epidemiologic knowledge of how that world behaves. There is strong evidence from the pre-IAP era and settings that don't implement IAP that cases don't occur at random among colonized women as the reviewer suggests but are much more likely to occur in certain subgroups. This in fact is the very basis of the risk-based approach to IAP used in many settings that does not rely on microbiologic screening for GBS colonization—because cases are much more likely to occur in infants born to mothers with these specific obstetric risk factors. Additionally, among colonized women, the preceding literature that motivated our study suggests that women with low rather than high antibody levels are those more at risk of having an infant with GBS disease. Thus, we feel the scenarios we assessed are more relevant approximations for the counterfactual world of interest and that a selection at random is not where the biology and epidemiology point us.
- With regards to the suggestion that thresholds from the counterfactual 'no IAP' analysis should be used for regulatory or other decision making, it is beyond the scope of our study to propose the immunologic endpoints for trials, merely to inform them; decision on the endpoints themselves is up to regulators and industry sponsors. However, we do stress here that our sensitivity analysis, while we find it helpful and reassuring, is indeed a counterfactual 'thought exercise' and we do not in the U.S. have the option to collect data directly on a population with no IAP.

Comment 3: One issue which warrants discussion at the least, is whether the selection of controls based on recto-vaginal colonization in pregnancy is a suitable "control" set for LOD cases, for which the pathogenesis of invasive GBS disease differ, and include possibility of acquisition from other than the mother (and even more so if such LOD cases are occurring beyond 3-4 weeks of life). It would be useful to provide information on the median age of the LOD cases enrolled. Although the current study does not permit for selection of any other set of controls, a discussion on this issue is warranted, including in what direction the SCRR might be influenced; and whether this could explain up to a 60 fold difference in SCRR observed for EOD compared with LOD.

Response: This reviewer makes a great point about the LOD controls—while they were selected to assure some exposure of the infant to GBS and this is accomplished most readily through use of mothers colonized with the same homotypic serotype as cases, it is correct that LOD epidemiology suggests that about half of LOD cases are born to mothers that are not colonized with GBS. While the reviewer proposes simply adding this as another limitation to the discussion, after further reflections we found a way to

design from within our study an additional late onset control group and now include this as an additional set of sensitivity analyses. Because the antibody binding assay is multiplex and generates results for all 6 serotypes, we have the option to create a new set of controls that includes some infants born to mothers colonized by the target serotype and some infants whose mothers were not colonized by the target serotype (these mothers were colonized by a non-target serotype). To this end, for the 4 serotypes that had sufficient case sample sizes for LOD curve generation, we randomly selected to retain three times the number of cases of the target serotype controls and added to this a random selection of the same number from the non-target controls (except for serotype III for which, due to sample size, we selected 50% of homotypic controls and the same number of heterotypic controls) and then re-ran the CALM models with this new control group. This had minimum effect on curve shape and associated thresholds and is now included as Figure 5 Panel G, along with the other sensitivity analyses.

Comment 4: It is a difficult to evaluate the stats methods as it has not yet been, consequently it is hard to check the code without more details of the model...

Response: We appreciate this and have now added a citation to the preprint that outlines the CALM methodology in detail. We also have now included a short appendix as a supplemental material to this study so that more details about CALM are readily accessible to readers. The CALM manuscript is in late-stage review and we hope to have an in press citation soon.

Specific comments:

Comment 5: Ln 39-40 change “yet to be established” to “yet to be approved by regulatory authorities”, as previous studies already have explored the same, even if using a different assay and for limited number of serotypes.

Response: The text was modified as suggested – see lines 40 and 90 in tracked changes version; lines 40 and 88 in clean version.

Comment 6: Ln 41- “change “near birth” and provide the window period - <XX days

Response: We prefer to use ‘near birth’ in the abstract given the 150-word limit, but details about the timing of blood spot collection are available in the text and Table 1.

Comment 7: Ln 46- see above comment re substituting “protection thresholds” to SCRR. The same applies throughout rest of the manuscript (albeit not highlighted further in the comments)

Response: We removed the term “correlates of protection” as suggested by reviewers. We also adopted the use of the Serological Thresholds associated with Risk Reduction (STRR) in response to reviewer comments. We use the STRR terminology when referring to the whole curve and range of thresholds but prefer to use ‘protective thresholds’ when referring to the three thresholds (75%, 80%, and 90%) that regulators have directly expressed interest in and above which cases are infrequent.

Comment 8: Ln 47- add “serological” before “endpoints”

Response: The text was modified as suggested.

Comment 9: Ln 105- considering that temporal changes have been observed in GBS recto-vaginal colonization, even if among the same dominant serotypes, the difference in time-period over which the cases (longer) and controls were enrolled could inadvertently influence the population sero-epidemiology. Although this could influence the results in either direction, it warrants some mentioning of in the discussion.

Response: It is true that the time period for cases is longer than for controls but we are not sure of the relevance because all our analyses were serotype-specific such that cases were all definitively exposed to the ‘target serotype’ for the analysis, and controls were all exposed to a mother colonized by that same target serotype. We are glad to address this concern of the reviewer’s more fully if we can understand the concern and hypothesis better.

Comment 10: Ln 116- suggest clarifying upfront whether controls were “enrolled”/identified matched for case serotype upfront, as opposed to random selection of controls (i.e. born to colonized women), and subsequent “{matching” to cases.

Response: There was no matching per se in this study, but we did conduct all the analyses stratified by ‘target serotype’ (the disease-causing serotype of the case or the maternal colonizing serotype the control was exposed to). Target serotype for controls was ascertained after enrollment as enrollment was the first step to collection of the maternal colonizing isolate which we then had to serotype in our study laboratory. It is for this reason that we could not control precisely the case to control ratio as can be observed in the study flow figure. The more common colonizing serotypes (in our study serotype II was most common) resulted in a much higher control to case ratio than the more rare colonizing serotypes such as serotype III.

Comment 11: Ln 137: Has the supporting evidence of correlation of serum IgG and DBS IgG been published? If so, it should be referenced upfront.

Response: The manuscript describing the correlation between serum and DBS is available on a preprint server (reference now included in the paper).

Comment 12: Ln 156: what is the sensitivity of real-time PCR on isolates compared with directly of selective broth. If higher on selective broth, could inadvertently be including “controls” with low levels of exposure (and consequently low risk of invasive disease than the general population where culture is used to establish risk of invasive disease. This could inadvertently result in inclusion of “controls” who had almost no risk for EOD, irrespective of maternal derived IgG levels. Suggest a sensitivity analysis whereby controls in whom serotype only detected by PCR are excluded.

Response: We are not sure we understand the rationale behind this comment—we are open to exploring it further with more clarification but based on our current understanding we propose no revisions beyond a clarification in the paragraph starting on line 158 that this refers to the methods used to serotype GBS colonization specimens from controls.

It is correct that EOD cases were identified by culture of a normally sterile site, most commonly blood. This however has no bearing on the density of colonization of the case mothers, which the reviewer, we think implies was more likely derived from culture than that of the controls? We don't have any evidence in our study data collection to support this hypothesis as we don't have validated data on the method by which GBS colonization screening was performed on cases nor do any of the results track directly with density of colonization. Additionally, we typically see in the era of widespread IAP in the U.S. that cases are enriched for mothers who were not screened or who had false negative results which can arise for a multiplicity of reasons. Finally, in the U.S. all recommended antenatal screening methods have an initial required enrichment step. We do not believe tests done post enrichment (be they culture or PCR) can shed reliable light on maternal colonization density. Thus, we feel we have no evidence in our collected data to suggest that controls were exposed to a very light colonization that should not 'count' as a GBS exposure.

Comment 13: Ln 192: unclear what “other factors” is being referred to.

Response: We removed cases and controls exposed to target serotypes since the central hypothesis in our study is that antibody levels to target serotypes may be higher in controls compared to cases. We felt participants exposed to non-target serotypes represented a cleaner group to focus on for the question of stability of antibody concentrations over different storage durations. We clarify this point in the text – see lines 206-210 in tracked changes version; 196-200 in clean version.

Comment 14: Ln 200-203: as this study involved newborn blood, assume that the collection of samples “after disease onset” will only apply to EOD casers. Suggest clarifying as such. Also, considering it would take 2-3 weeks for an IgG response to materialise after infection, the issue would be possibility of waning of IgG or lowering due to adsorption by invasive isolates. Suggest clarifying.

Response: Thank you for this suggestion. We added language to clarify that this issue is most relevant to EOD cases in both the manuscript and the sensitivity analysis figure – see line 300 in tracked changes version; line 288 in clean version.

Comment 15: Ln 274- does collection within 3 days only apply to EOD?

Response: 25/28 of cases with blood spot collection >72 hours after disease onset were EOD cases. DBS collection after the onset of LOD may take place if there was a delay until an unstable or critically ill neonate became stable, or if the DBS collection was repeated due to inconclusive newborn screening test results. Clarifying language was added to the manuscript (see line 300 in tracked changes version; line 288 in clean version) and sensitivity analysis (Figure 5, panel B).

Comment 16: Ln 278-280: considering the highlight dynamic nature of GBS colonization, even with clearance and new acquisitions within a 4 week period, it is possible that some controls were no longer exposed to GBS at birth. Also, earlier recto-vaginal acquisition could elicit IgG responses higher than GBS acquisition closer to the time of delivery. This warrants discussion under study limitations.

Response: Thanks for raising this. We do conduct a sensitivity analysis that excludes controls born to mothers screened greater than 5 weeks before delivery because that group has a reasonable probability not to have been exposed to GBS at the time of birth. This sensitivity analysis yielded results similar to analyses including such women. Based on this we doubt that colonization losses within the 5 weeks preceding delivery, while these may occur, will influence the study findings since loss of colonization within this time window is more rare; similarly gains of colonization during that window do also occur but rarely—this evidence in fact forms the basis of the recommended window for late antenatal screening for GBS. Thus, we feel this concern of the reviewer’s, while valid, is likely minor and given the more important limitations already addressed in the Discussion we don’t feel this warrants attention unless the Editors so request.

Comment 17: Line 277 - unclear if this was done in the analysis or this was a sensitivity analysis

Response: This was done as a sensitivity analysis. The results were similar to the main results. We have modified the sensitivity analysis figure to make it easier for the reader to track all the analyses conducted.

Comment 18. Line 289 - Are counterfactual cases informative if they are randomly drawn from those with antibody concentrations below the LLOQ, and subsequently are BLQ? Isn't it only cases and control above the LLOQ that are informative to disease risk curve?

Response: See the response to this reviewer's comment 2 above for an expanded explanation of the rationale behind the counterfactual scenarios we evaluated. We designed these based on the known epidemiology of GBS and IAP and then assessed the resulting curves. While one scenario was based on flipping those with low antibody levels, and the reviewer is correct that this may have less influence on curve shape, this scenario was anchored in the published evidence that low anti-GBS CPS IgG levels are associated with increased infant disease risk—in fact it is this prior evidence that motivated our current study. The other counterfactual scenario was based solely on flipping controls whose mothers presented in labor with specific clinical risk factors and thus was completely agnostic to antiCPS IgG values. Both scenarios produced curves and thresholds similar to those in our primary analysis, likely due to the small number of expected counterfactual cases.

Comment 19: Ln 289-292- see comment 2 above

Response: Yes, please see our response to comment 2 above.

Comment 20: Ln 307- an incorrect reference is being used

Response: This has been corrected.

Comment 21: Ln 385- the second sentence seems to be incomplete

Response: A typo in this sentence was corrected.

Comment 22: Ln 408-409. Suggest removing reference to attempt at developing maternal vaccines against Klebs and E. coli, where most disease occur in preterm

babies, and more complexities as to whether or not maternal vaccination would be possible to reduce the risk.

Response: Reference to *Klebsiella pneumoniae* and *Escherichia coli* has been removed – see line 469 in tracked changes version; line 466 in clean version.

Comment 23: Ln 409- requires referencing

Response: The phrase in question has been removed – see line 470 in tracked changes version.

Comment 24: Fig 2B- interestingly, there is variability in % of cases and controls with undetectable antibody. Notably, this percentage is lowest for serotype II, which coincidentally is also the serotype with the highest SCRR. Also, serotype II is a relatively uncommon colonizing serotype. Is there any hypothesis that could explain this observation, including the possibility of serotype II IgG being cross-reactive to antigen from another bacterium? Also, this highlights the issue of exercising caution when using SCRR as a proxy for “correlates of protection”.

Response: We also find it interesting that serotype II has relatively high proportions of cases and controls with quantifiable antibody concentrations and relatively high protective thresholds. In contrast to what the reviewer states, serotype II is actually the most common colonizing serotype (see figure 1) and the 3rd most common cause of invasive infant disease behind serotypes III and Ia. As with our response to comment 9 from reviewer #1 - we understand the reviewer’s interests to understand the causal mechanisms underlying these observations; however, exploration of these mechanisms was beyond the scope of our study.

Reviewer #2 (Remarks on code availability): Unable to review the "code" as the statistical methods is lacking in detail

Response: We have now cited the preprint for the CALM method and included a brief appendix on the CALM method in the supplementary material. We trust this now helps with review of the code.

Reviewer #3 (Remarks to the Author):

The authors present a large case-control study assessing the correlation between newborn anti-CPS IgG antibody levels and the risk of invasive Group B Streptococcus (GBS) disease, aiming to establish correlates of protection against early- vs late-onset GBS disease (EOD vs LOD). Such thresholds would assist in applying an immunogenic rather than disease prevention outcome for Phase III studies of maternal GBS vaccines. These thresholds were investigated in prior work, but as the authors note, serotype-specific thresholds were not investigated or nor were differences in correlates of protection against early vs late-onset disease (EOD and LOD).

The study enrolled 643 cases and 2802 controls across multiple U.S. states, utilizing serotype-specific antibody concentration data from dried blood spots collected within 48 hrs of birth. Findings suggest that while higher antibody concentrations are protective against GBS in general, serotype- and age-at-onset-specific thresholds vary substantially, with lower thresholds required for LOD versus EOD.

Strengths include a relatively large sample size allowing the key subgroup analyses, the use of standardized serologic assays, and adjustment for some relevant covariates. Additionally, the authors performed extensive sensitivity analyses to assess a subset of potential confounders (IAP, timing of screening, timing of DBS in relationship to disease onset, race and ethnicity) indicating that curves are generally similar in these subgroups. The authors state that the use of dried blood spots collected shortly after birth should minimize the potential influence of antibody decay or post-infection immune responses on antibody concentrations. This study would be challenging for other groups to perform given the extensive multi-state nature of the consortium; in that sense, it is highly original.

The writing is clear; the abstract, introduction and conclusions are generally very clear and easy to understand.

Response: Thank you for these appreciative comments on what was a challenging but rewarding study!

There are several notable limitations to be addressed:

Comment 1: Some concern remains about correlation between cord blood sera or early neonatal sera levels of antibody and DBS antibody levels. The authors address this in lines 167-173 of the manuscript, stating, “The concordance correlation coefficient (CCC) for a panel of 33 matched sera and DBS demonstrated accuracy and precision ≥ 0.98 for each serotype...” but then go on to discuss the need to apply “conversion factors...for serotypes II, III, and IV (29)”, based on Deming regression results. The cited paper (reference 29) is not available for review and does not even appear to be in press, the

citation says the manuscript is “in progress”. Additional detail is needed to assure the reader that DBS antibody levels are equivalent or sufficiently equivalent to those in cord blood or early (e.g. 1st 48 hrs of life) neonatal sera for all serotypes or at least the primary disease-causing serotypes, as the paper touts the establishment of correlates of protection across serotypes that can be used as endpoints in Phase III vaccine trials as a main strength of the paper. Thus, we must be assured that these endpoints are robust and accurate and I think some additional information in this regard (as presumably would appear in reference 29 if it were available) would be helpful.

Response: See the first 2 responses to the Methods-related comments from Reviewer 1. The DBS to serum bridging manuscript is now available on a preprint server so all the details are accessible and submitted to a peer reviewed journal. We apologize that this citation was not included with the original submissions as it is very fair to want to see the details. Also in response to Reviewer 1’s comments we added some clarifications about the bridging work to the text.

Comment 2: Could the authors provide more clarification on the use of DBS within first 48 hrs of life and those antibody levels to predict protection from late-onset GBS disease that could occur very distant from that antibody sample. How much variability was there in timing of late onset disease and did antibody decay rates play a factor/how was that accounted for?

Response: We have clarified in the manuscript (introduction, second paragraph) that the GBS vaccine field has coalesced around immunologic endpoints for infant GBS disease based on newborn blood at or as close to birth as possible. Thus, while waning of antibody levels will occur for late onset cases, particularly given the median onset at 31 days of life (now added to Table 1), this waning is not relevant for ascertainment of the serologic threshold (at birth) of risk reduction. Additionally, the median age of late onset cases in our study and the distribution around it matches that in our broader population-based active surveillance population allowing for generalizability to U.S. late onset cases. Certainly among late onset cases, those with onset at day 7 of life may have a different serologic threshold of risk reduction from late onset cases with onset at age 89 of life but no practical study design would allow for a sufficient sample size to estimate different STRRs by day at onset nor would such STRRs be practical in the regulatory setting. Thus, we have used the traditional early vs late onset stratification that has long been of utility in the field.

Comment 3: differences in case and control enrollment periods: The cases were enrolled from 2010 to 2022, whereas controls were collected from 2018 to 2022, which

may introduce bias due to potential changes (i.e., secular trends) in GBS epidemiology, maternal immunity, or clinical management over time.

Response: We agree these enrollment periods differ and we raise this at the start of the limitations paragraph in the Discussion. Our primary concern about how this time difference could introduce bias was the differential in DBS storage times before processing between cases and controls; our cross-sectional assessment of antibody concentrations with different storage times was reassuring in this regard. While we agree other secular trends in GBS epidemiology may have occurred over the longer time period of case enrollment we are not sure we understand by what hypothesis such changes could affect our results. As stated in response to Reviewer 2, comment 9, as part of our study design both cases and controls were exposed to the ‘target serotype’ of GBS under evaluation—cases were infected by the serotype and controls were born to mothers colonized by the serotype. It is hard to picture how secular trends in GBS circulating strains or clinical management would affect this group of infants exposed to a particular known serotype of GBS, particularly as apart from IAP which we do address, there are no other clinical management strategies that would prevent an infant from getting GBS disease—although clinical management improvements may have led to better outcomes of infants with GBS disease during this time period.

Comment 4: variation in specimen storage conditions: Some dried blood spot specimens were stored for extended periods, and although sensitivity analyses suggest minimal degradation effects, long-term storage may still impact antibody stability. (This concern is compounded by the difference in enrollment periods, as noted above).

Response: Again, this is a limitation we already raise in the Discussion section and we addressed by conducting a cross sectional assessment of stability of our analytes that did not suggest an effect of long term storage, particularly from 2013 forward. We appreciate the reviewer is listing in this review the limitations we already have flagged and discussed; if the Editor feels there is anything further that needs to be added to the text we will be glad to address this.

Comment 5: lack of functional antibody data: The study relies on antibody concentrations without directly assessing functional protection (e.g., via opsonophagocytic killing assays), which could provide additional biologically-relevant information.

Response: Again we raise this point already in the Discussion at the end of the fourth paragraph. Functional antibody results are anticipated to play only a supportive role to IgG-based STRR for a number of reasons including that OPK assays have much more variability than the IgG MIA assay reported here, and that a very high proportion of case

and control samples contain antibiotics which can affect the assay performance. We feel the existing text in the Discussion that states such data when available may provide 'additional insights' is sufficient but if the Editors would like more details we are glad to accommodate.

Comment 6: limited generalizability to low-resource settings: The study population is primarily from high-income countries with widespread intrapartum antibiotic prophylaxis (IAP), limiting direct applicability to settings with greater disease burden, different patterns of antibiotic use and potentially different serotype distributions.

Response: We agree with this point and one of the contributions of our study was to provide data from a high income/widespread IAP setting in contrast to published South Africa and Finland data. We added a sentence to this effect in paragraph 5 of the discussion as well at the end of the limitations paragraph in the Discussion, in response to a comment from Reviewer 2 – see line 495-7 in tracked changes version; line 472 in clean version). Geography does appear to influence baseline population antibody levels and circulating GBS serotypes (although given the small number of GBS serotypes overall there is much overlap between settings)—the extent to which geography influences STRR remains to be seen and the CALM method we employed here is suited to such meta-analyses that could treat study location as a model covariate.

Comment 7: potential influence of IAP on control selection: Because most controls received IAP, some infants who might have developed EOD without antibiotics were classified as disease-free, possibly skewing the estimated protective thresholds. Appreciate that a sensitivity analysis was performed to include some of the controls as cases, but still this should be clearly acknowledged as a limitation.

Response: We agree and have added a sentence to the limitations paragraph of the Discussion to call this out for EOD – see lines 513-515 in tracked changes version; line 489 in clean version.

Comment 8: serotype-specific sample size constraints: The study did not have sufficient cases to generate curves for some serotypes (e.g., 1b and V for LOD), leading to gaps in understanding serotype-specific protection despite the authors' noting this as a key distinction from prior work.

Response: We agree and had already called this out in the Discussion in paragraph 5 (Limitations).

Comment 9: uncertainty in protective threshold estimation: The protective thresholds

vary significantly by serotype and disease onset, but confidence intervals are large in some cases, making precise vaccine target-setting challenging.

Response: We agree and have expanded our callout to this in the Discussion paragraph 5 – limitations - line 522 in tracked changes version; line 497 in clean version.

Comment 10: in light of case/control differences highlighted in Figure 1 consort diagram (e.g., in terms of loss of samples), further explicit contrasts of the samples lost in the two groups would help the reader determine whether these mismatches merit concern.

Response: Contrast of the samples lost are provided below. We do not think differences between cases and controls impacted the study findings for the reasons listed in the table below. We prefer to keep Figure 1 as is, since it is already quite detailed and the numbers are available to readers interested to make these comparisons; however, we could include these comparisons if required.

	Cases N=792	Controls N=3,367	Comment
Serotype not available or not Ia-V	80 (10.1%)	228 (6.8%)	We do not think this difference merits concern as the reasons for these differences (see 2 rows below) are unlikely to be associated with IgG concentration
Isolate not available	76 (9.6%)	64 (1.9%)	This difference likely reflects the population-based nature of ABCs surveillance in which complete (100%) case capture is expected regardless of isolate availability.
Isolate not serotypes Ia-V	4 (0.5%)	164 (4.9%)	This difference was expected as the serotype distribution for GBS colonization is more diverse than the serotype distribution for invasive GBS disease.
Assay result (IgG concentration) not available	45 (5.7%)	326 (9.7%)	This difference is largely due to:  • The large number (n=286) of serotype II control DBS that were not processed as we already had >10 serotype-matched controls per case. After careful consideration, we decided that these additional serotype II controls would be marginally informative and the additional data did not justify the time and resources required to run the assay. • One study site (MD) had a higher proportion of DBS that were in poor condition and/or produced light colored eluent. This site contributed a sizable number of cases, but few controls.
Blood transfusion prior to blood spot collection	24 (3.0%)	11 (0.3%)	This difference was expected as preterm infants are at higher risk of invasive GBS disease (more likely to be cases) and more likely to require blood transfusion. Inclusion of infants with blood transfusion prior to blood spot collection complicates interpretation of the IgG

			concentration available to provide protection near birth.
--	--	--	---

Comment 11: Figure 2A and B should indicate the number of samples per group; particularly where the titer is 0, it is difficult to see how many samples contribute.

Response: The sample sizes are in the caption, but we would be open to suggestions for how to add into the figure themselves.

Comment 12: Figure 4 could be more clearly presented in tabular form.

Response: We have updated Figure 4 to present the thresholds as both a figure (4a) and table (4b).

Comment 13: Figure 5, the crucial sensitivity analyses, should probably also be presented grouped by serotype (rather than parameter tested) in a supplemental figure.

Response: If any of the factors explored through the sensitivity analyses were true confounders or otherwise impacted the findings, we would expect to see consistent differences across serotypes. With that in mind, we prefer the current layout that allows the reader to easily assess the impact of each factor across serotype; however, we can reformat if required.

Comment: Authors have appropriately cited relevant prior work, notably the phase II study by Mahdi (NEJM 2023) and the work by Gilbert et al. (Vaccine 2022) outlining standards for establishing correlates of protection for GBS vaccines.

Response: We agree these are important, relevant publications.

Reviewer #3 (Remarks on code availability): I did not assess the code.

REVIEWER COMMENTS

Reviewer #2 (Remarks to the Author):

The authors have adequately responded to the majority of the previous Reviewers comments and made the necessary edits in the manuscript. Also, the additional analysis undertaken, including in relation to LOD is appreciated. Remaining issues that warrant further consideration are:

Response: Thank you for your comments, which have greatly improved the manuscript! We have provided additional responses below.

1. Reviewer 2 , Comment 2 response: Whilst agreeing that IAP is not a confounder, as would not affect antibody concentration, there still is an inherent bias to only have included controls with titres >LLQ in the sensitivity analysis. As evident from the study, even among cases, there is a wide dispersion of IgG, including some cases with high titres. Similarly, at the population level- one would expect a wide dispersion of titers among controls, and it biases toward a higher STRR by only including controls with detectable titres. I would request that the sensitivity analysis include a random selection of controls, agnostic to antibody titres.

Response: We have included an additional sensitivity analysis including a random selection of controls in Figure 5E. The results are in line with the primary analyses.

2. Re authors response to Reviewer 2 , Comment 2 – it is correct that there are risk factors for invasive GBS, with maternal GBS colonization being the major risk factor for EOD. However, a fair proportion of cases do still occur randomly, as risk factors are not identified in all cases even in the presence of IAP. If that was not the case, the rationale for selection of controls as done in studies such as this would be questionable- as controls will be inherently at no/low risk of EOD to follow through with the argument of cases not occurring randomly. Also, whilst its correct as shown in the current and earlier studies, that serotype specific IgG is inversely related to risk of invasive, this does not detract from the population heterogeneity (more so in controls) of antibody titres in case and controls. As such, disagree that controls should be only chose from those with measurable antibody titres, as opposed to random selection.

Response: We have included an additional sensitivity analysis including a random selection of controls in Figure 5E. The results are in line with the primary analyses.

3. Rev 2 Comment 8 (and Rev 3 comment 3)- Both reviewers highlight the difference in time-period over which cases and controls were enrolled. Whilst the authors agree that there could be temporal changes in colonization, the effect thereof on temporal changes in antibody concentrations is not considered. As maternal colonization induces antibody responses, the temporal changes in serotype-specific colonization would also be associated with temporal changes in antibody titers. Consequently, the potential effect thereof warrants discussion, as its beyond the study framework to be able to account for such (other than another sensitivity analysis limiting controls to being temporally linked to cases).

Response: We have included a statement in the limitations section of the discussion to address this issue.

Reviewer #4 (Remarks to the Author):

The authors performed a large sero-epidemiologic study of GBS disease across eight sites in the United States. The authors analysed the data with a variety of appropriate statistical methods. Their manuscript was rigorously reviewed by several reviewers, and the authors responded to all of the reviewers' suggestions for improvements, making the majority of the suggested changes and clarifications, and respectfully defending their position for the remainder. The review process has further improved the overall analyses and narrative of this study. This report of their findings is likely to prove highly valuable in the ongoing assessment and future implementation of GBS vaccines.

Response: Thank you for your review!